# Compositional Law Parsing with Latent Random Functions

**Fan Shi    Bin Li**[*]  **Xiangyang Xue**
Shanghai Key Laboratory of Intelligent Information Processing
School of Computer Science, Fudan University
`fshi22@m.fudan.edu.cn   {libin,xyxue}@fudan.edu.cn`

## Abstract

Human cognition has compositionality. We understand a scene by decomposing the scene into different concepts (e.g., shape and position of an object) and learning the respective laws of these concepts, which may be either natural (e.g., laws of motion) or man-made (e.g., laws of a game). The automatic parsing of these laws indicates the model's ability to understand the scene, which makes law parsing play a central role in many visual tasks. This paper proposes a deep latent variable model for Compositional LAw Parsing (CLAP), which achieves the human-like compositionality ability through an encoding-decoding architecture to represent concepts in the scene as latent variables. CLAP employs concept-specific latent random functions instantiated with Neural Processes to capture the law of concepts. Our experimental results demonstrate that CLAP outperforms the baseline methods in multiple visual tasks such as intuitive physics, abstract visual reasoning, and scene representation. The law manipulation experiments illustrate CLAP's interpretability by modifying specific latent random functions on samples. For example, CLAP learns the laws of position-changing and appearance constancy from the moving balls in a scene, making it possible to exchange laws between samples or compose existing laws into novel laws.

## 1 Introduction

Compositionality is an important feature of human cognition (Lake et al., 2017). Humans can decompose a scene into individual concepts to learn the respective laws of these concepts, which can be either natural (e.g. laws of motion) or man-made (e.g. laws of a game). When observing a scene of a moving ball, one tends to parse the changing patterns of its appearance and position separately: The appearance stays consistent over time, while the position changes according to the laws of motion. By composing the laws of the ball's appearance and position, one can understand the changing pattern and predict the status of the moving ball.

Although compositionality has inspired a number of models in visual understanding such as representing handwritten characters through hierarchical decomposition of characters (Lake et al., 2011; 2015) and representing a multi-object scene with object-centric representations (Eslami et al., 2016; Kosiorek et al., 2018; Greff et al., 2019; Locatello et al., 2020), automatic parsing of laws in a scene is still a great challenge. For example, to understand the rules in abstract visual reasoning such as Raven's Progressive Matrices (RPMs) test, the comprehension of attribute-specific representations and the underlying relationships among them is crucial for a model to predict the missing images (Santoro et al., 2018; Steenbrugge et al., 2018; Wu et al., 2020). To understand the laws of motion in intuitive physics, a model needs to grasp the changing patterns of different attributes (e.g. appearance and position) of each object in a scene to predict the future (Agrawal et al., 2016; Kubricht et al., 2017; Ye et al., 2018). However, these methods usually employ neural networks to directly model changing patterns of the scene in black-box fashion, and can hardly abstract the laws of individual concepts in an explicit, interpretable and even manipulatable way.

A possible solution to enable the abovementioned ability for a model is exploiting a function to represent a law of concept in terms of the representation of the concept itself. To represent a law

---
[*]Corresponding author

that may depict arbitrary changing patterns, an expressive and flexible random function is required. The Gaussian Process (GP) is a classical family of random functions that achieves the diversity of function space through different kernel functions (Williams & Rasmussen, 2006). Recently proposed random functions (Garnelo et al., 2018b; Eslami et al., 2018; Kumar et al., 2018; Garnelo et al., 2018a; Singh et al., 2019; Kim et al., 2019; Louizos et al., 2019; Lee et al., 2020; Foong et al., 2020) describe function spaces with the powerful nonlinear fitting ability of neural networks. These random functions have been used to capture the changing patterns in a scene, such as mapping timestamps to frames to describe the physical law of moving objects in a video (Kumar et al., 2018; Singh et al., 2019; Fortuin et al., 2020). However, these applications of random functions take images as inputs and the captured laws account for all pixels instead of expected individual concepts.

In this paper, we propose a deep latent variable model for Compositional LAw Parsing (CLAP) [1]. CLAP achieves the human-like compositionality ability through an encoding-decoding architecture (Hamrick et al., 2018) to represent concepts in the scene as latent variables, and further employ concept-specific random functions in the latent space to capture the law on each concept. By means of the plug-in of different random functions, CLAP gains generality and flexibility applicable to various law parsing tasks. We introduce CLAP-NP as an example that instantiates latent random functions with recently proposed Neural Processes (Garnelo et al., 2018b).

Our experimental results demonstrate that the proposed CLAP outperforms the compared baseline methods in multiple visual tasks including intuitive physics, abstract visual reasoning, and scene representation. In addition, the experiment on exchanging latent random functions on a specific concept and the experiment on composing latent random functions of the given samples to generate new samples both well demonstrate the *interpretability* and even *manipulability* of the proposed method for compositional law parsing.

## 2 RELATED WORK

**Compositional Scene Representation** Compositional scene representation models (Yuan et al., 2022a) can understand a scene through object-centric representations (Yuan et al., 2019b;a; 2021; Emami et al., 2021; Yuan et al., 2022b). Some models initialize and update object-centric representations through iterative computational processes like neural expectation maximization (Greff et al., 2017), iterative amortized inference (Greff et al., 2019), and iterative cross-attention (Locatello et al., 2020). Other models adopt non-iterative computational processes to parse disentangled attributes for objects as latent variables (Eslami et al., 2016) or extract representations from evenly divided regions in parallel (Lin et al., 2020b). Recently, many models focus on capturing layouts of scenes (Jiang & Ahn, 2020) or learning consistent object-centric representations from videos (Kosiorek et al., 2018; Jiang et al., 2019; Lin et al., 2020a). Unlike CLAP, compositional scene representation models learn how objects in a scene are composed but cannot explicitly represent underlying laws and understand how these laws constitute the changing pattern of the scene.

**Random Functions** The GP (Williams & Rasmussen, 2006) is a classical family of random functions that regards the outputs of a function as a random variable of multivariate Gaussian distribution. To incorporate neural networks with random functions, some models encode functions through global representations (Wu et al., 2018; Eslami et al., 2018; Gordon et al., 2019). NP (Garnelo et al., 2018b) captures function stochasticity with a Gaussian distributed latent variable. According to the way of stochasticity modeling, one can construct random functions with different characteristics (Kim et al., 2019; Louizos et al., 2019; Lee et al., 2020; Foong et al., 2020). And other models develop random functions by learning adaptive kernels (Tossou et al., 2019; Patacchiola et al., 2020) or computing the integral of ODEs or SDEs on latent states (Norcliffe et al., 2021; Li et al., 2020; Hasan et al., 2021). Random functions provide explicit representations for laws but cannot model the compositionality of laws.

## 3 PRELIMINARIES

A random function describes a distribution over function space $\mathcal{F}$, from which we can sample a function $f$ mapping the input sequence $(\boldsymbol{x}_1, ..., \boldsymbol{x}_N)$ to output sequence $(\boldsymbol{y}_1, ..., \boldsymbol{y}_N)$. The impor-

---

[1]Code is available at https://github.com/FudanVI/generative-abstract-reasoning/tree/main/clap

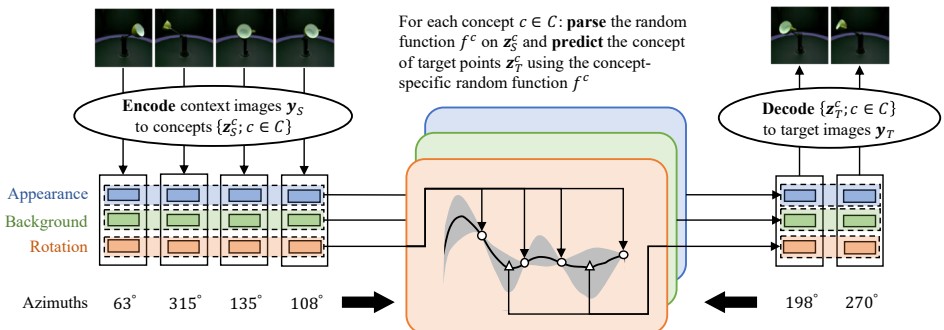

Figure 1: Overview of CLAP: to predict target images, CLAP first encodes the given context images into representations of concepts, such as object appearance, background and angle of object rotation; then parses concept-specific latent random functions and computes the representations of concepts in the target images; finally decodes these concepts to compose the target images.

tant role of a random function is uncovering the underlying function over some given points and predicting function values at novel positions accordingly. In detail, given a set of context points $D_S = \{(\boldsymbol{x}_s, \boldsymbol{y}_s) | s \in S\}$, we should find the most possible function that $\boldsymbol{y}_s = f(\boldsymbol{x}_s)$ for these context points, then use $f(\boldsymbol{x}_t)$ to predict $\boldsymbol{y}_t$ for target points $D_T = \{(\boldsymbol{x}_t, \boldsymbol{y}_t) | t \in T\}$. The abovementioned prediction problem can be regarded as estimating a conditional probability $p(\boldsymbol{y}_T | \boldsymbol{x}_T, D_S)$.

**Neural Process (NP)**   NP (Garnelo et al., 2018b) captures function stochasticity with a Gaussian distributed latent variable $\boldsymbol{g}$ and lets $p(\boldsymbol{y}_T | \boldsymbol{x}_T, D_S) = \int \prod_{t \in T} p(\boldsymbol{y}_t | \boldsymbol{g}, \boldsymbol{x}_t) q(\boldsymbol{g} | D_S) d\boldsymbol{g}$. In this case, $p(\boldsymbol{y}_T | \boldsymbol{x}_T, D_S)$ consists of a generative process $p(\boldsymbol{y}_t | \boldsymbol{g}, \boldsymbol{x}_t)$ and an inference process $q(\boldsymbol{g} | D_S)$, which can be jointly optimized through variational inference. To estimate $p(\boldsymbol{y}_T | \boldsymbol{x}_T, D_S)$, NP extracts the feature of each context point in $D_S$ and sum them up to acquire the mean $\boldsymbol{\mu}$ and deviation $\boldsymbol{\sigma}$ of $\boldsymbol{g}$:

$$\boldsymbol{\mu}, \boldsymbol{\sigma} = \mathcal{T}_f \left( \frac{1}{|S|} \sum_{s \in S} \mathcal{T}_a \left( \boldsymbol{y}_s, \boldsymbol{x}_s \right) \right). \tag{1}$$

Then NP samples the global latent variable $\boldsymbol{g}$ from $q(\boldsymbol{g} | D_S)$. NP can independently predict $\boldsymbol{y}_t$ at the target position $\boldsymbol{x}_t$ through the generative process $p(\boldsymbol{y}_t | \boldsymbol{g}, \boldsymbol{x}_t) = \mathcal{N}(\boldsymbol{y}_t | \mathcal{T}_p(\boldsymbol{g}, \boldsymbol{x}_t), \sigma_y^2 \boldsymbol{I})$. Neural networks $\mathcal{T}_a$, $\mathcal{T}_f$ and $\mathcal{T}_p$ increase the model capacity of NP to learn data-specific function spaces.

## 4  MODEL

Figure 1 is the overview of CLAP. The encoder converts context images to individual concepts in the latent space. Then CLAP parses concept-specific random functions on the concept representations of context images (context concepts) to predict the concept representations of target images (target concepts). Finally, a decoder maps the target concepts to target images. The concept-specific latent random functions achieve the compositional law parsing in CLAP. We will introduce the generative process, variational inference, and parameter learning of CLAP. At the end of this section, we propose CLAP-NP that instantiates the latent random function with the Neural Process (NP).

### 4.1  LATENT RANDOM FUNCTIONS

Given $D_S$ and $D_T$, the objective of CLAP is maximizing $\log p(\boldsymbol{y}_T | \boldsymbol{y}_S, \boldsymbol{x})$. In the framework of conditional latent variable models, we introduce a variational posterior $q(\boldsymbol{h} | \boldsymbol{y}, \boldsymbol{x})$ for the latent variable $\boldsymbol{h}$ and approximate the log-likelihood with evidence lower bound (ELBO) (Sohn et al., 2015):

$$\log p\left(\boldsymbol{y}_T | \boldsymbol{y}_S, \boldsymbol{x}\right) \geq \mathbb{E}_{q(\boldsymbol{h}|\boldsymbol{y},\boldsymbol{x})} \big[ \log p\left(\boldsymbol{y}_T | \boldsymbol{h}\right) \big] - \mathbb{E}_{q(\boldsymbol{h}|\boldsymbol{y},\boldsymbol{x})} \left[ \log \frac{q\left(\boldsymbol{h}|\boldsymbol{y},\boldsymbol{x}\right)}{p\left(\boldsymbol{h}|\boldsymbol{y}_S,\boldsymbol{x}\right)} \right] = \mathcal{L}. \tag{2}$$

The latent variable $\boldsymbol{h}$ includes the global latent random functions $f$ and local information of images $\boldsymbol{z} = \{\boldsymbol{z}_n\}_{n=1}^N$ where $\boldsymbol{z}_n$ is the low-dimensional representation of $\boldsymbol{y}_n$. Moreover, we decompose $\boldsymbol{z}_n$ into independent concepts $\{\boldsymbol{z}_n^c | c \in C\}$ (e.g., an image of a single-object scene may have concepts

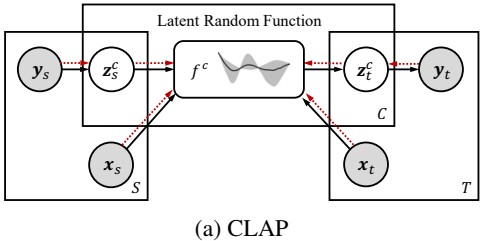
(a) CLAP

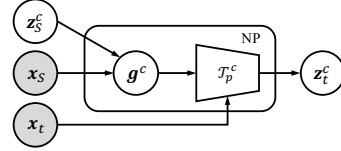
(b) NP as the latent random function

Figure 2: The graphical model of CLAP where the generative process is indicated using black solid lines and the variational inference is indicated using red dotted lines. Panel (a) shows the framework of CLAP, where the latent random function $f^c$ for the concept $c \in C$ can be instantiated by arbitrary random functions; Panel (b) describes how to instantiate the latent random function with NP.

such as object appearance, illumination, camera orientation, etc.) where $C$ refers to the name set of these concepts. Assuming that these concepts satisfy their respective changing patterns, we employ independent latent random functions to capture the law of each concept. Denote the latent random function on the concept $c$ as $f^c$, the specific form of $f^c$ depends on the way we model it. As Figure 2b shows, if adopting NP as the latent random function, we have $f^c(\boldsymbol{x}_t) = \mathcal{T}_p^c(\boldsymbol{g}^c, \boldsymbol{x}_t)$ where $\boldsymbol{g}^c$ is a Gaussian-distributed latent variable to control the randomness of $f^c$. Within the graphical model of CLAP in Figure 2a, the prior and posterior in $\mathcal{L}$ are factorized into

$$p(\boldsymbol{h}|\boldsymbol{y}_S, \boldsymbol{x}) = \prod_{c \in C} \left( p\left(f^c|\boldsymbol{z}_S^c, \boldsymbol{x}_S\right) \prod_{t \in T} p\left(\boldsymbol{z}_t^c|f^c, \boldsymbol{z}_S^c, \boldsymbol{x}_S, \boldsymbol{x}_t\right) \prod_{s \in S} p\left(\boldsymbol{z}_s^c|\boldsymbol{y}_s\right) \right)$$

$$p\left(\boldsymbol{y}_T|\boldsymbol{h}\right) = \prod_{t \in T} p\left(\boldsymbol{y}_t|\boldsymbol{z}_t\right), \quad q(\boldsymbol{h}|\boldsymbol{y}, \boldsymbol{x}) = \prod_{c \in C} \left( q(f^c|\boldsymbol{z}^c, \boldsymbol{x}) \prod_{s \in S} q(\boldsymbol{z}_s^c|\boldsymbol{y}_s) \prod_{t \in T} q(\boldsymbol{z}_t^c|\boldsymbol{y}_t) \right) \tag{3}$$

Decomposing $f$ into concept-specific laws $\{f^c|c \in C\}$ in the above process is critical for CLAP's compositionality. The compositionality makes it possible to parse only several concept-specific laws rather than the entire law on $\boldsymbol{z}$, which reduces the complexity of law parsing while increasing the interpretability of laws. The interpretability allows us to manipulate laws, for example, exchanging the law on some concepts and composing the laws of existing samples.

## 4.2 PARAMETER LEARNING

Based on Equations 2 and 3, we factorize the ELBO as (See Appendix B.3 for the detailed derivation)

$$\mathcal{L} = \underbrace{\sum_{t \in T} \mathbb{E}_{q(\boldsymbol{h}|\boldsymbol{y}, \boldsymbol{x})} \left[ \log p\left(\boldsymbol{y}_t|\boldsymbol{z}_t\right) \right]}_{\text{Reconstruction term } \mathcal{L}_r} - \underbrace{\sum_{c \in C} \sum_{t \in T} \mathbb{E}_{q(\boldsymbol{h}|\boldsymbol{y}, \boldsymbol{x})} \left[ \log \frac{q(\boldsymbol{z}_t^c|\boldsymbol{y}_t)}{p\left(\boldsymbol{z}_t^c|f^c, \boldsymbol{z}_S^c, \boldsymbol{x}_S, \boldsymbol{x}_t\right)} \right]}_{\text{Target regularizer } \mathcal{L}_t}$$

$$- \underbrace{\sum_{c \in C} \sum_{s \in S} \mathbb{E}_{q(\boldsymbol{h}|\boldsymbol{y}, \boldsymbol{x})} \left[ \log \frac{q(\boldsymbol{z}_s^c|\boldsymbol{y}_s)}{p\left(\boldsymbol{z}_s^c|\boldsymbol{y}_s\right)} \right]}_{\text{Context regularizer } \mathcal{L}_s} - \underbrace{\sum_{c \in C} \mathbb{E}_{q(\boldsymbol{h}|\boldsymbol{y}, \boldsymbol{x})} \left[ \log \frac{q(f^c|\boldsymbol{z}^c, \boldsymbol{x})}{p\left(f^c|\boldsymbol{z}_S^c, \boldsymbol{x}_S\right)} \right]}_{\text{Function regularizer } \mathcal{L}_f} \tag{4}$$

**Reconstruction Term**  $\mathcal{L}_r$ is the sum of $\log p(\boldsymbol{y}_t|\boldsymbol{z}_t)$, which is modeled with a decoder to reconstruct target images from the representation of concepts. CLAP maximizes $\mathcal{L}_r$ to connect the latent space with the image space, enabling the decoder to generate high-quality images.

**Target Regularizor**  $\mathcal{L}_t$ consists of Kullback–Leibler (KL) divergences between the posterior $q(\boldsymbol{z}_t^c|\boldsymbol{y}_t)$ and the prior $p(\boldsymbol{z}_t^c|f^c, \boldsymbol{z}_S^c, \boldsymbol{x}_S, \boldsymbol{x}_t)$ of target concepts. The parameter of $q(\boldsymbol{z}_t^c|\boldsymbol{y}_t)$ is directly computed using the encoder to contain more accurate information about the target image while $p(\boldsymbol{z}_t^c|f^c, \boldsymbol{z}_S^c, \boldsymbol{x}_S, \boldsymbol{x}_t)$ is estimated from the context. The minimization of $\mathcal{L}_t$ fills the gap between the posterior and prior to ensure the accuracy of predictors.

**Context Regularizor**  $\mathcal{L}_s$ consists of KL divergences between the posterior $q(\boldsymbol{z}_s^c|\boldsymbol{y}_s)$ and the prior $p(\boldsymbol{z}_s^c|\boldsymbol{y}_s)$ of context concepts. To avoid mismatch between the posterior and prior and reduce model

parameters, the posterior and prior are parameterized with the same encoder. In this case, we have $p(\boldsymbol{z}_s^c|\boldsymbol{y}_s) = q(\boldsymbol{z}_s^c|\boldsymbol{y}_s)$ and further $\mathcal{L}_s = 0$. So we remove $\mathcal{L}_s$ from Equation 4.

**Function Regularizor** $\mathcal{L}_f$ consists of KL divergences between the posterior $q(f^c|\boldsymbol{z}^c, \boldsymbol{x})$ and the prior $p(f^c|\boldsymbol{z}_S^c, \boldsymbol{x}_S)$. We use $\mathcal{L}_f$ as a measure of function consistency to ensure that model can obtain similar distributions for $f^c$ with any subset $\boldsymbol{z}_S^c \subseteq \boldsymbol{z}^c$ and $\boldsymbol{x}_S \subseteq \boldsymbol{x}$ as inputs. To this end, CLAP shares concept-specific function parsers to model $q(f^c|\boldsymbol{z}^c, \boldsymbol{x})$ and $p(f^c|\boldsymbol{z}_S^c, \boldsymbol{x}_S)$. And we need to design the architecture of function parsers so that they can take different subsets as input.

Correct decomposition of concepts is essential for model performance. However, CLAP is trained without additional supervision to guide the learning of concepts. Although CLAP explicitly defines independent concepts in priors and posteriors, it is not sufficient to automatically decompose concepts for some complex data. To solve the problem, we can introduce extra inductive biases to promote concept decomposition by designing proper network architectures, e.g., using spatial transformation layers to decompose spatial and non-spatial concepts (Skafte & Hauberg, 2019). We can also set hyperparameters to control the regularizers (Higgins et al., 2017), or add a total correlation (TC) regularizer $\mathcal{R}_{TC}$ to help concept learning (Chen et al., 2018). Here we choose to add extra regularizers to the ELBO of CLAP, and finally, the optimization objective becomes

$$\underset{p,q}{\arg\max}\, \mathcal{L}^* = \underset{p,q}{\arg\max}\, \mathcal{L}_r - \beta_t \mathcal{L}_t - \beta_f \mathcal{L}_f - \beta_{TC}\mathcal{R}_{TC} \tag{5}$$

where $\beta_t$, $\beta_f$, $\beta_{TC}$ are hyperparameters that regulate the importance of regularizers in the training process. We compute $\mathcal{L}^*$ through Stochastic Gradient Variational Bayes (SGVB) estimator (Kingma & Welling, 2013) and update parameters using gradient descent optimizers (Kingma & Ba, 2015).

### 4.3 NP as Latent Random Function

In this subsection, we propose CLAP-NP as an example that instantiates the latent random function with NP. According to NP, we employ a latent variable $\boldsymbol{g}^c$ to control the randomness of the latent random function $f^c$ for each concept in CLAP-NP.

**Generative Process** CLAP-NP first encodes each context image $\boldsymbol{y}_s$ to the concepts:

$$\begin{aligned} \{\boldsymbol{\mu}_{z,s}^c|c \in C\} &= \text{Encoder}(\boldsymbol{y}_s), & s \in S, \\ \boldsymbol{z}_s^c &\sim \mathcal{N}\left(\boldsymbol{\mu}_{z,s}^c, \sigma_z^2 \boldsymbol{I}\right), & c \in C, s \in S. \end{aligned} \tag{6}$$

To stabilize the learning of concepts, the encoder outputs only the mean of Gaussian distribution, with the standard deviation as a hyperparameter. Using the process in Equation 1 for each concept, a concept-specific function parser aggregates and transforms the contextual information to the mean $\boldsymbol{\mu}_g^c$ and standard deviation $\boldsymbol{\sigma}_g^c$ of $\boldsymbol{g}^c$. As Figure 2b shows, the concept-specific target predictor $\mathcal{T}_p^c$ takes $\boldsymbol{g}^c \sim \mathcal{N}(\boldsymbol{\mu}_g^c, \text{diag}(\boldsymbol{\sigma}_g^2))$ and $\boldsymbol{x}_t$ as inputs to predict the mean $\boldsymbol{\mu}_{z,t}^c$ of $\boldsymbol{z}_t$, leaving the standard deviation as a hyperparameter. To keep independence between concepts, CLAP-NP introduces identical but independent function parsers and target predictors. Once all of the concepts $\boldsymbol{z}_t^c \sim \mathcal{N}(\boldsymbol{\mu}_{z,t}^c, \sigma_z^2 \boldsymbol{I})$ for $c \in C$ are generated, we concatenate and decode them into target images:

$$\boldsymbol{y}_t \sim \mathcal{N}\left(\boldsymbol{\mu}_{y,t}, \sigma_y^2 \boldsymbol{I}\right), \quad \boldsymbol{\mu}_{y,t} = \text{Decoder}\left(\{\boldsymbol{z}_t^c|c \in C\}\right), \quad t \in T. \tag{7}$$

To control the noise in sampled images, we introduce a hyperparameter $\sigma_y$ as the standard deviation. As Figure 2a shows, we can rewrite the prior $p(\boldsymbol{h}|\boldsymbol{y}_S, \boldsymbol{x})$ in CLAP-NP as

$$p(\boldsymbol{h}|\boldsymbol{y}_S, \boldsymbol{x}) = \prod_{c \in C} \left( p\left(\boldsymbol{g}^c|\boldsymbol{z}_S^c, \boldsymbol{x}_S\right) \prod_{t \in T} p\left(\boldsymbol{z}_t^c|\boldsymbol{g}^c, \boldsymbol{x}_t\right) \prod_{s \in S} p\left(\boldsymbol{z}_s^c|\boldsymbol{y}_s\right) \right). \tag{8}$$

**Inference and Learning** In the variational inference, other than that the randomness of $f^c$ is replaced by $\boldsymbol{g}^c$, the posterior of CLAP-NP is the same as that in Equation 3:

$$q(\boldsymbol{h}|\boldsymbol{y}, \boldsymbol{x}) = \prod_{c \in C} \left( q(\boldsymbol{g}^c|\boldsymbol{z}^c, \boldsymbol{x}) \prod_{s \in S} q(\boldsymbol{z}_s^c|\boldsymbol{y}_s) \prod_{t \in T} q(\boldsymbol{z}_t^c|\boldsymbol{y}_t) \right). \tag{9}$$

In the first stage, we compute the means of both $\boldsymbol{z}_S$ and $\boldsymbol{z}_T$ using the encoder in CLAP. Because the encoder is shared in the prior and posterior, we obtain $\boldsymbol{z}_s^c$ through Equation 6 instead of recalculating

Table 1: MSEs on BoBa, CRPM, and MPI3D dataset. The training configurations of datasets are given in the brackets, and the test configurations are displayed in the table headers.

| Model | BoBa-1 ($\eta = 1/12 \sim 4/12$) | | BoBa-2 ($\eta = 1/12 \sim 4/12$) | | CRPM-DT ($\eta = 1/9 \sim 2/9$) | |
|---|---|---|---|---|---|---|
| | $\eta = 4/12$ | $\eta = 6/12$ | $\eta = 4/12$ | $\eta = 6/12$ | $\eta = 2/9$ | $\eta = 3/9$ |
| NP | $420.7 \pm 4.4$ | $649.2 \pm 6.0$ | $1169.6 \pm 17.4$ | $1917.8 \pm 24.8$ | $59.3 \pm 4.7$ | $101.9 \pm 12.2$ |
| GP | $1165.5 \pm 78.4$ | $2064.9 \pm 76.6$ | $2062.6 \pm 137.1$ | $3556.8 \pm 132.8$ | $263.2 \pm 19.0$ | $425.3 \pm 22.0$ |
| GQN | $844.7 \pm 36.6$ | $1915.9 \pm 34.1$ | $1512.2 \pm 22.9$ | $3104.5 \pm 40.9$ | $178.3 \pm 9.2$ | $379.3 \pm 8.0$ |
| CLAP-NP | $\mathbf{70.7 \pm 2.7}$ | $\mathbf{151.2 \pm 11.1}$ | $\mathbf{912.0 \pm 52.4}$ | $\mathbf{1833.8 \pm 91.9}$ | $\mathbf{19.6 \pm 0.7}$ | $\mathbf{49.5 \pm 10.7}$ |

| Model | CRPM-DC ($\eta = 1/9 \sim 2/9$) | | MPI3D ($\eta = 1/8 \sim 4/8$) | | | |
|---|---|---|---|---|---|---|
| | $\eta = 2/9$ | $\eta = 3/9$ | $\eta = 4/8$ | $\eta = 6/8$ | $\eta = 10/40$ | $\eta = 20/40$ |
| NP | $192.6 \pm 13.1$ | $323.2 \pm 13.6$ | $131.5 \pm 3.9$ | $230.5 \pm 4.0$ | $274.9 \pm 4.2$ | $562.8 \pm 6.2$ |
| GP | $468.3 \pm 40.6$ | $764.5 \pm 39.0$ | $268.9 \pm 13.0$ | $491.6 \pm 12.9$ | $279.3 \pm 8.2$ | $650.4 \pm 12.5$ |
| GQN | $414.3 \pm 7.2$ | $823.0 \pm 15.3$ | $\mathbf{55.6 \pm 0.7}$ | $331.8 \pm 2.8$ | $772.1 \pm 3.4$ | $1148.9 \pm 3.2$ |
| CLAP-NP | $\mathbf{119.5 \pm 17.2}$ | $\mathbf{264.5 \pm 27.9}$ | $58.8 \pm 2.3$ | $\mathbf{153.2 \pm 6.3}$ | $\mathbf{105.5 \pm 0.6}$ | $\mathbf{213.9 \pm 1.0}$ |

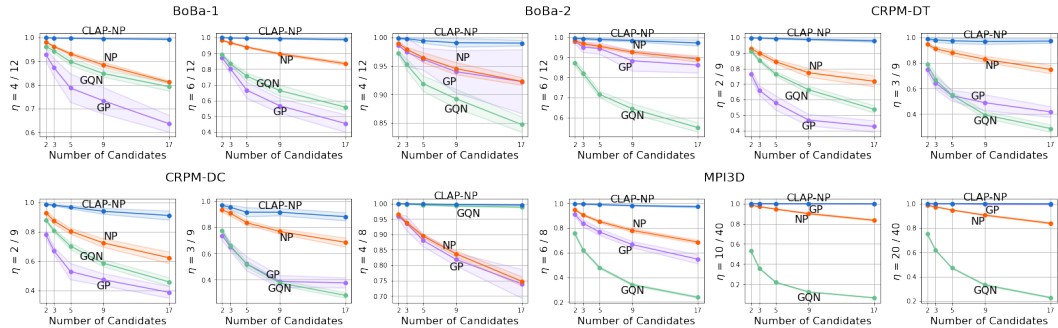

Figure 3: SA scores on BoBa, CRPM, and MPI3D dataset where the blue, orange, purple, and green lines denote scores of CLAP-NP, NP, GP, and GQN.

$z_s \sim q(z_s^c|y_s)$. Then we compute the distribution parameters of $g^c$ through the shared concept-specific function parsers and the input $z^c$ and $x$. With the above generative and inference processes, some subterms in the ELBO of CLAP-NP become

$$\mathcal{L}_t = \sum_{c \in C} \sum_{t \in T} \mathbb{E}_{q(h|y,x)} \left[ \log \frac{q(z_t^c|y_t)}{p(z_t^c|g^c, x_t)} \right], \quad \mathcal{L}_f = \sum_{c \in C} \mathbb{E}_{q(h|y,x)} \left[ \log \frac{q(g^c|z^c, x)}{p(g^c|z_S^c, x_S)} \right]. \quad (10)$$

## 5 EXPERIMENTS

To evaluate the model's ability of compositional law parsing on different types of data, we use three datasets in the experiments: (1) **Bouncing Ball** (abbreviated as **BoBa**) dataset (Lin et al., 2020a) to validate the ability of intuitive physics; (2) **Continuous Raven's Progressive Matrix (CRPM)** dataset (Shi et al., 2021) to validate the ability of abstract visual reasoning; (3) **MPI3D** dataset (Gondal et al., 2019) to validate the ability of scene representation. We adopt NP (Garnelo et al., 2018b), GP with the deep kernel (Wilson et al., 2016), and GQN (Eslami et al., 2018) as baselines. We consider NP and GP as baselines because they are representative random functions that construct the function space in different ways, and CLAP-NP instantiates the latent random function with NP. GQN is adopted since it models the distribution of image-valued functions, which highly fits our experimental settings. NP is not used to model image-valued functions, so we add an encoder and decoder to NP's aggregator and predictor to help it handle high-dimensional images. For GP, we use a pretrained encoder and decoder to reduce the dimension of raw images. See Appendix D.1 and D.2 for a detailed introduction to datasets, hyperparameters, and architectures of models.

For quantitative experiments, we adopt Mean Squared Error (MSE) and Selection Accuracy (SA) as evaluation metrics. Since non-compositional law parsing will increase the risk of producing undesired changes on constant concepts and further cause pixel deviations in predictions, we adopt MSE as an indicator of prediction accuracy. To compute the concept-level deviations between predictions

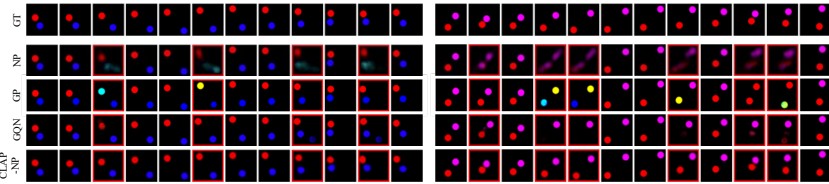

Figure 4: Prediction results (in red boxes) on BoBa-2 with $\eta = 4/12$ (left) and $\eta = 6/12$ (right).

Figure 5: Prediction results (in red boxes) of CRPM-DT with $\eta = 2/9$ (top) and $\eta = 3/9$ (bottom). We provide two predictions of the same context to illustrate a special case: when a row of images is removed from the given matrix, all predictions with the progressive law of size and constant law of color are correct. In this special case, only CLAP-NP captures such prediction uncertainty.

and ground truths, which we think can be an indicator of semantic error, we introduce SA as another metric to calculate the distance on the representation space. Since the scale of representation-level distances is tightly according to the representation space of models, we turn to get SA by selecting target images from a set of candidates. For $(\boldsymbol{x}, \boldsymbol{y})$, we combine the target images $\boldsymbol{y}_T$ with $K$ targets randomly selected from other samples to establish a candidate set. Then the candidates and the prediction $\tilde{\boldsymbol{y}}_T \sim p(\boldsymbol{y}_T|\boldsymbol{y}_S, \boldsymbol{x})$ of the models are encoded into representations to calculate concept-level distances. Finally, the candidate with the smallest distance is chosen, and SA is the selection accuracy across all test samples. In the following experiments, we use $\eta = N_T/N$ to denote the training or test configuration where $N_T$ of $N$ images in the sample are target images to predict.

## 5.1 INTUITIVE PHYSICS

To evaluate the intuitive physics of models, which is the underlying knowledge for humans to understand the evolution of the physical world (Kubricht et al., 2017), we use BoBa dataset where physical laws of collisions and motions are described as functions mapping timestamps to frames. BoBa-1 and BoBa-2 are one-ball and two-ball instances of BoBa. When predicting target frames, models should learn both constant laws of appearance and physical laws of collisions. We set up two test configurations $\eta = 4/12$ and $\eta = 6/12$ for BoBa. The quantitative results in Table 1 and Figure 3 illustrate CLAP-NP's intuitive physics on both BoBa-1 and BoBa-2. Figure 4 visualizes the predictions of BoBa-2 where NP can only generate blurred images while GP and GQN can hardly understand the appearance constancy of balls. That is, balls in the predictions of GP have different colors, and GQN generates non-existent balls and ignores existing balls. Instead, CLAP-NP can separately capture the appearance constancy and collision laws, which benefits the law parsing in scenes with novel appearances and physical laws (Appendix E.7).

## 5.2 ABSTRACT VISUAL REASONING

Abstract visual reasoning is a manifestation of human's fluid intelligence (Cattell, 1963). And RPM (Raven & Court, 1998) is a famous nonverbal test widely used to estimate the model's abstract visual reasoning ability. RPM is a $3 \times 3$ image matrix, and the changing rule of the entire matrix consists of multiple attribute-specific subrules. In this experiment, we evaluate models on four instances of the CRPM dataset and two test configurations $\eta = 2/9$ and $\eta = 3/9$. In Table 1 and Figure 3, CLAP-NP achieves the best results with the metrics in both MSE and SA. Figure 5 displays the predictions on CRPM-DT. In the results, NP produces ambiguous-edged outer triangles, GP predicts targets with incorrect laws, and GQN generates multiple similar images. It is worth stressing that the answer to

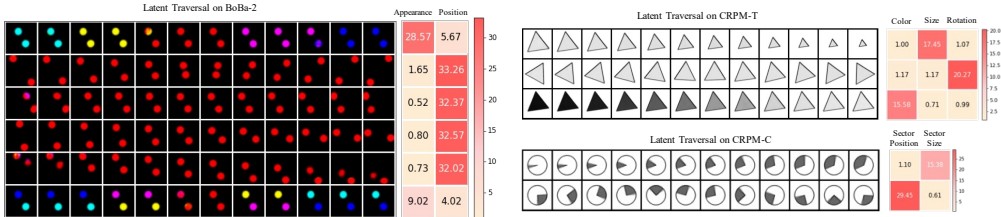

Figure 6: The qualitative latent traversal results and quantitative variance declines on BoBa-2, CRPM-T, and CRPM-C. High variance declines indicate significant correlations between laws and LRFs, by which we can determine whether a dimension encodes the law we want to edit.

an RPM is not unique when we remove the row from it. We display examples of this situation in the second row of Figure 5, indicating that CLAP-NP can understand progressive and constant laws on different attributes rather than simple context-to-target mappings.

## 5.3 SCENE REPRESENTATION

In scene representation, we use MPI3D dataset where a robotic arm in scenes manipulates the altitude and azimuth of the object to produce images with different laws. The laws of scenes can be regarded as functions that project object altitudes or azimuths to RGB images. In this task, the key to predicting correct target images is to determine which law the scene follows by observing given context images. We train models under $\eta = 1/8 \sim 4/8$ and test them with more target or scene images ($\eta = 4/8, 6/8, 10/40, 20/40$). The MSE and SA scores in Table 1 and Figure 3 demonstrate that CLAP-NP outperforms NP in all configurations and GQN in $\eta = 6/8, 10/40, 20/40$. CLAP-NP has a slightly higher SA but lower MSE scores than GQN when $\eta = 4/8$. A possible reason is that the network of GQN has good inductive biases and more parameters to generate clearer images and achieve better MSE scores. However, such improvement in the pixel-level reconstruction has limited influence on the representation space, as well as the SA scores of GQN. GQN can hardly generalize the random functions to novel situations with more images, leading to performance loss when $\eta = 6/8, 10/40, 20/40$. GP's MSE scores on MPI3D indicate that it is suitable for scenes with more context images (e.g., $\eta = 10/40$). With only sparse points, the prediction uncertainty of GP leads to a significant decrease in performance. However, compositionality allows CLAP-NP to parse concept-specific subrules, which improves the model's generalization ability in novel test configurations.

## 5.4 COMPOSITIONALITY OF LAWS

It is possible to exactly manipulate a law if one knows which latent random functions (LRFs) encode it. We adopt two methods to find the correspondence between laws and LRFs. Figure 6 visualizes the changing patterns by perturbing and decoding concepts, enabling us to understand the meaning of LRFs. Another way is to compute variance declines between laws and LRFs (Kim & Mnih, 2018). Assume that we have ground truth laws $L$ and a generator to produce sample batches by fixing the arbitrary law $l \in L$. First, we generate a batch of samples without fixing any laws and parse LRFs for these samples to estimate the variance $\bar{v}^c$ of each concept-specific LRF. By fixing the law $l$, we generate some batches $\{\mathcal{B}_l^b\}_{b=1}^{N_B}$ and estimate the in-batch variance $\{v_l^{c,b}; c \in C\}$ of LRFs for $\mathcal{B}_l^b$. Finally, the variance decline between the law $l$ and the concept $c$ is

$$s_{l,c} = \frac{1}{N_B} \sum_{b=1}^{N_B} \frac{\bar{v}^c - v_l^{c,b}}{\bar{v}^c}. \tag{11}$$

In Figure 6, we display the variance declines on BoBa-2, CRPM-T, and CRPM-C, which will guide the manipulation of laws in the following experiments. And Figure 7 visualizes the law manipulation process, which well illustrates the motivation of our model on compositional law parsing.

**Latent Random Function Exchange**    The top of Figure 7 shows how we swap the laws of samples with the aid of compositional law parsing. To exchange the law of appearance between samples, we first refer to the variance declines in Figure 6. For BoBa-2, the laws are represented with 6 LRFs

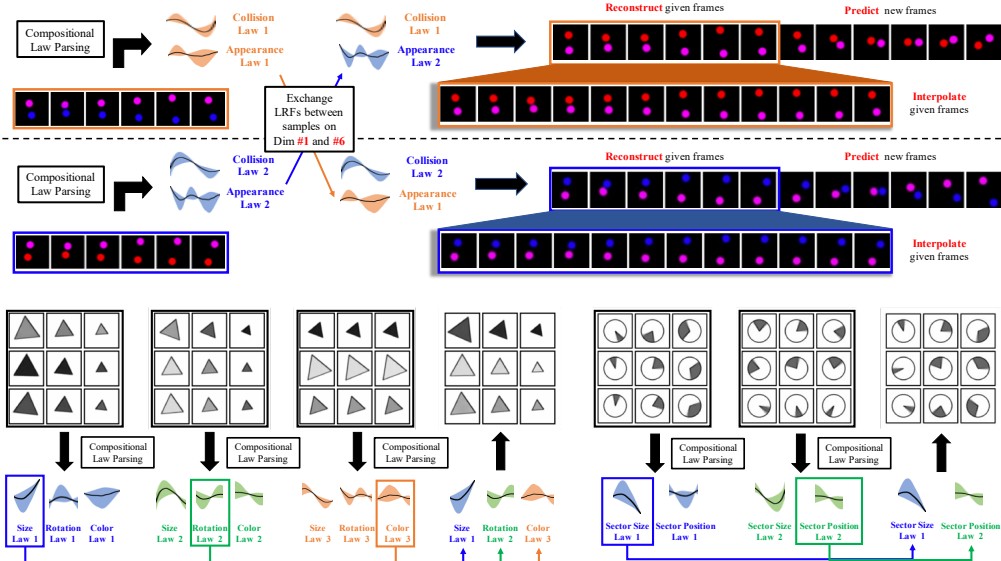

Figure 7: An illustration of law manipulation. At the top, we exchange the law of appearance by swapping the LRFs of two samples on the first and last dimensions. At the bottom, we compose laws from existing samples of CRPM-T and CRPM-C to generate novel samples.

where the first and last LRFs encode the law of appearance while the others indicate the motion of balls. Thus we infer the LRFs (global rule latent variables in CLAP-NP) of two samples and swap the LRFs of two samples on the first and last dimensions. Finally, we regenerate the edited samples using the generative process, and further, make predictions and interpolations on them.

**Latent Random Function Composition**    For samples of CRPM, we manipulate laws by combining the LRFs from existing samples to generate samples with novel laws. Figure 7 shows the results of law composition on CRPM-T. The first step is parsing LRFs for given samples. By querying the variance declines in Figure 6, we can find these LRFs respectively correspond to the laws of size, rotation, and color. Since we have figured out the relations between laws and LRFs, the next step is to pick laws from three different samples respectively and compose them into an unseen combination of laws. Finally, the composed laws are decoded to generate a sample with novel changing patterns. This process is similar to the way a human designs RPMs (i.e., set up separate sub-laws for each attribute and combine them into a complete PRM). From another perspective, law composition provides an option for us to generate novel samples from existing samples (Lake et al., 2015).

## 6    CONCLUSION AND LIMITATIONS

Inspired by the compositionality in human cognition, we propose a deep latent variable model for Compositional LAw Parsing (CLAP). CLAP decomposes high-dimensional images into independent visual concepts in the latent space and employs latent random functions to capture the concept-changing laws, by which CLAP achieves the compositionality of laws. To instantiate CLAP, we propose CLAP-NP that uses NPs as latent random functions. The experimental results demonstrate the benefits of our model in intuitive physics, abstract visual reasoning, and scene representation. Through the experiments on latent random function exchange and composition, we further qualitatively evaluate the interpretability and manipulability of the laws learned by CLAP.

**Limitations.** (1) **Complexity of datasets.** Because compositional law parsing is an underexplored task, we first validate the effectiveness of CLAP on datasets with relatively clear and simple laws to avoid the influence of unknown confounders in complicated datasets. (2) **Setting the number of LRFs.** For scenes with multiple complex laws, we can empirically set an appropriate upper bound or directly put a large upper bound on the number of LRFs. However, using too many LRFs increases model parameters, and the redundant concepts decrease CLAP's computational efficiency. See Appendix F for detailed limitations and future works.

ACKNOWLEDGMENTS

This work was supported in part by the National Natural Science Foundation of China (No.62176060), STCSM projects (No.20511100400, No.22511105000), Shanghai Municipal Science and Technology Major Project (2021SHZDZX0103), and the Program for Professor of Special Appointment (Eastern Scholar) at Shanghai Institutions of Higher Learning.

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

## A   APPENDIX

In Supplementary Materials, we (1) provide the details of ELBO, then (2) introduce the datasets, model architecture, hyperparameters, and computational resources adopted in our experiments, and finally (3) provide additional experimental results. In particular, we provide additional results of *editing* and *manipulating* latent random functions, which validate our motivation and contribution.

## B   DETAILS OF ELBO

### B.1   ELBO OF CONDITIONAL LATENT VARIABLE MODELS

$$
\begin{aligned}
\log p\left(\boldsymbol{y}_T|\boldsymbol{y}_S,\boldsymbol{x}\right) &= \int_{\boldsymbol{h}} q\left(\boldsymbol{h}|\boldsymbol{y},\boldsymbol{x}\right)\log p\left(\boldsymbol{y}_T|\boldsymbol{y}_S,\boldsymbol{x}\right)d\boldsymbol{h} \\
&= \int_{\boldsymbol{h}} q\left(\boldsymbol{h}|\boldsymbol{y},\boldsymbol{x}\right)\log\frac{p\left(\boldsymbol{h},\boldsymbol{y}_T|\boldsymbol{y}_S,\boldsymbol{x}\right)}{p\left(\boldsymbol{h}|\boldsymbol{y},\boldsymbol{x}\right)}d\boldsymbol{h} \\
&= \int_{\boldsymbol{h}} q\left(\boldsymbol{h}|\boldsymbol{y},\boldsymbol{x}\right)\log\frac{p\left(\boldsymbol{h},\boldsymbol{y}_T|\boldsymbol{y}_S,\boldsymbol{x}\right)}{q\left(\boldsymbol{h}|\boldsymbol{y},\boldsymbol{x}\right)}d\boldsymbol{h} + \int_{\boldsymbol{h}} q\left(\boldsymbol{h}|\boldsymbol{y},\boldsymbol{x}\right)\log\frac{q\left(\boldsymbol{h}|\boldsymbol{y},\boldsymbol{x}\right)}{p\left(\boldsymbol{h}|\boldsymbol{y},\boldsymbol{x}\right)}d\boldsymbol{h} \quad (12) \\
&\geq \int_{\boldsymbol{h}} q\left(\boldsymbol{h}|\boldsymbol{y},\boldsymbol{x}\right)\log\frac{p\left(\boldsymbol{h},\boldsymbol{y}_T|\boldsymbol{y}_S,\boldsymbol{x}\right)}{q\left(\boldsymbol{h}|\boldsymbol{y},\boldsymbol{x}\right)}d\boldsymbol{h} \\
&= \mathbb{E}_{q(\boldsymbol{h}|\boldsymbol{y},\boldsymbol{x})}\left[\log p\left(\boldsymbol{y}_T|\boldsymbol{h}\right)\right] - \mathbb{E}_{q(\boldsymbol{h}|\boldsymbol{y},\boldsymbol{x})}\left[\log\frac{q\left(\boldsymbol{h}|\boldsymbol{y},\boldsymbol{x}\right)}{p\left(\boldsymbol{h}|\boldsymbol{y}_S,\boldsymbol{x}\right)}\right] = \mathcal{L}.
\end{aligned}
$$

### B.2   PRIOR AND POSTERIOR FACTORIZATION

$$
\begin{aligned}
p\left(\boldsymbol{y}_T|\boldsymbol{h}\right) &= \prod_{t\in T} p\left(\boldsymbol{y}_t|\boldsymbol{z}_t\right) \\
p(\boldsymbol{h}|\boldsymbol{y}_S,\boldsymbol{x}) &= \prod_{c\in C} p\left(f^c,\boldsymbol{z}_S^c,\boldsymbol{z}_T^c|\boldsymbol{y}_S,\boldsymbol{x}\right) \\
&= \prod_{c\in C} p\left(\boldsymbol{z}_T^c|f^c,\boldsymbol{z}_S^c,\boldsymbol{x}_S,\boldsymbol{x}_T\right)p\left(f^c|\boldsymbol{z}_S^c,\boldsymbol{x}_S\right)p\left(\boldsymbol{z}_S^c|\boldsymbol{y}_S\right) \\
&= \prod_{c\in C}\left(p\left(f^c|\boldsymbol{z}_S^c,\boldsymbol{x}_S\right)\prod_{t\in T} p\left(\boldsymbol{z}_t^c|f^c,\boldsymbol{z}_S^c,\boldsymbol{x}_S,\boldsymbol{x}_t\right)\prod_{s\in S} p\left(\boldsymbol{z}_s^c|\boldsymbol{y}_s\right)\right) \quad (13) \\
q(\boldsymbol{h}|\boldsymbol{y},\boldsymbol{x}) &= \prod_{c\in C} q(f^c,\boldsymbol{z}_S^c,\boldsymbol{z}_T^c|\boldsymbol{y},\boldsymbol{x}) \\
&= \prod_{c\in C} q(f^c|\boldsymbol{z}^c,\boldsymbol{x})q(\boldsymbol{z}_S^c|\boldsymbol{y}_S)q(\boldsymbol{z}_T^c|\boldsymbol{y}_T) \\
&= \prod_{c\in C}\left(q(f^c|\boldsymbol{z}^c,\boldsymbol{x})\prod_{s\in S} q(\boldsymbol{z}_s^c|\boldsymbol{y}_s)\prod_{t\in T} q(\boldsymbol{z}_t^c|\boldsymbol{y}_t)\right)
\end{aligned}
$$

### B.3   ELBO OF CLAP

$$
\begin{aligned}
\mathcal{L} = {}& \mathbb{E}_{q(\boldsymbol{h}|\boldsymbol{y},\boldsymbol{x})}\left[\log\prod_{t\in T} p\left(\boldsymbol{y}_t|\boldsymbol{z}_t\right)\right] \\
&- \mathbb{E}_{q(\boldsymbol{h}|\boldsymbol{y},\boldsymbol{x})}\left[\log\frac{\prod_{c\in C}\left(q(f^c|\boldsymbol{z}^c,\boldsymbol{x})\prod_{s\in S} q(\boldsymbol{z}_s^c|\boldsymbol{y}_s)\prod_{t\in T} q(\boldsymbol{z}_t^c|\boldsymbol{y}_t)\right)}{\prod_{c\in C}\left(\prod_{t\in T} p\left(\boldsymbol{z}_t^c|f^c,\boldsymbol{z}_S^c,\boldsymbol{x}_S,\boldsymbol{x}_t\right)p\left(f^c|\boldsymbol{z}_S^c,\boldsymbol{x}_S\right)\prod_{s\in S} p\left(\boldsymbol{z}_s^c|\boldsymbol{y}_s\right)\right)}\right] \quad (14)
\end{aligned}
$$

$$
\begin{aligned}
&= \sum_{t \in T} \mathbb{E}_{q(\boldsymbol{h}|\boldsymbol{y},\boldsymbol{x})} \big[ \log p\left(\boldsymbol{y}_t | \boldsymbol{z}_t\right) \big] - \sum_{c \in C} \mathbb{E}_{q(\boldsymbol{h}|\boldsymbol{y},\boldsymbol{x})} \left[ \log \frac{\prod_{t \in T} q(\boldsymbol{z}_t^c | \boldsymbol{y}_t)}{\prod_{t \in T} p\left(\boldsymbol{z}_t^c | f^c, \boldsymbol{z}_S^c, \boldsymbol{x}_S, \boldsymbol{x}_t\right)} \right] \\
&\quad - \sum_{c \in C} \mathbb{E}_{q(\boldsymbol{h}|\boldsymbol{y},\boldsymbol{x})} \left[ \log \frac{q(f^c | \boldsymbol{z}^c, \boldsymbol{x})}{p\left(f^c | \boldsymbol{z}_S^c, \boldsymbol{x}_S\right)} \right] - \sum_{c \in C} \mathbb{E}_{q(\boldsymbol{h}|\boldsymbol{y},\boldsymbol{x})} \left[ \log \frac{\prod_{s \in S} q(\boldsymbol{z}_s^c | \boldsymbol{y}_s)}{\prod_{s \in S} p\left(\boldsymbol{z}_s^c | \boldsymbol{y}_s\right)} \right] \\
&= \underbrace{\sum_{t \in T} \mathbb{E}_{q(\boldsymbol{h}|\boldsymbol{y},\boldsymbol{x})} \big[ \log p\left(\boldsymbol{y}_t | \boldsymbol{z}_t\right) \big]}_{\text{Reconstruction term } \mathcal{L}_r} - \underbrace{\sum_{c \in C} \sum_{t \in T} \mathbb{E}_{q(\boldsymbol{h}|\boldsymbol{y},\boldsymbol{x})} \left[ \log \frac{q(\boldsymbol{z}_t^c | \boldsymbol{y}_t)}{p\left(\boldsymbol{z}_t^c | f^c, \boldsymbol{z}_S^c, \boldsymbol{x}_S, \boldsymbol{x}_t\right)} \right]}_{\text{Target regularizer } \mathcal{L}_t} \\
&\quad - \underbrace{\sum_{c \in C} \sum_{s \in S} \mathbb{E}_{q(\boldsymbol{h}|\boldsymbol{y},\boldsymbol{x})} \left[ \log \frac{q(\boldsymbol{z}_s^c | \boldsymbol{y}_s)}{p\left(\boldsymbol{z}_s^c | \boldsymbol{y}_s\right)} \right]}_{\text{Context regularizer } \mathcal{L}_s} - \underbrace{\sum_{c \in C} \mathbb{E}_{q(\boldsymbol{h}|\boldsymbol{y},\boldsymbol{x})} \left[ \log \frac{q(f^c | \boldsymbol{z}^c, \boldsymbol{x})}{p\left(f^c | \boldsymbol{z}_S^c, \boldsymbol{x}_S\right)} \right]}_{\text{Function regularizer } \mathcal{L}_f}
\end{aligned}
$$

### B.3.1 RECONSTRUCTION TERM

$$
\begin{aligned}
\mathcal{L}_r &= \sum_{t \in T} \mathbb{E}_{q(\boldsymbol{h}|\boldsymbol{y},\boldsymbol{x})} \big[ \log p\left(\boldsymbol{y}_t | \boldsymbol{z}_t\right) \big] = \sum_{t \in T} \mathbb{E}_{q(\boldsymbol{z}_t|\boldsymbol{y}_t)} \big[ \log p\left(\boldsymbol{y}_t | \boldsymbol{z}_t\right) \big] \\
&\approx \sum_{t \in T} \log p\left(\boldsymbol{y}_t | \tilde{\boldsymbol{z}}_t\right), \quad \text{where } \tilde{\boldsymbol{z}}_t \sim q(\boldsymbol{z}_t | \boldsymbol{y}_t)
\end{aligned}
\tag{15}
$$

### B.3.2 TARGET REGULARIZER

$$
\begin{aligned}
\mathcal{L}_t &= \sum_{c \in C} \sum_{t \in T} \mathbb{E}_{q(\boldsymbol{h}|\boldsymbol{y},\boldsymbol{x})} \left[ \log \frac{q(\boldsymbol{z}_t^c | \boldsymbol{y}_t)}{p\left(\boldsymbol{z}_t^c | f^c, \boldsymbol{z}_S^c, \boldsymbol{x}_S, \boldsymbol{x}_t\right)} \right] \\
&= \sum_{c \in C} \sum_{t \in T} \mathbb{E}_{q(\boldsymbol{z}_S^c | \boldsymbol{y}_S)} \left[ \mathbb{E}_{q(\boldsymbol{z}_T^c | \boldsymbol{y}_T)} \left[ \mathbb{E}_{q(f^c | \boldsymbol{z}^c, \boldsymbol{x})} \left[ \log \frac{q(\boldsymbol{z}_t^c | \boldsymbol{y}_t)}{p\left(\boldsymbol{z}_t^c | f^c, \boldsymbol{z}_S^c, \boldsymbol{x}_S, \boldsymbol{x}_t\right)} \right] \right] \right].
\end{aligned}
\tag{16}
$$

Because of the function consistency in samples, function posteriors $q(f^c | \tilde{\boldsymbol{z}}^c, \boldsymbol{x})$ derived from arbitrary sampled $\tilde{\boldsymbol{z}}^c \sim q(\boldsymbol{z}^c | \boldsymbol{y})$ need to be consistent. So we let $f^c$ condition on only the observations $\boldsymbol{x}$ and $\boldsymbol{y}$ to simplify the posterior distribution, that is, we replace $q(f^c | \tilde{\boldsymbol{z}}^c, \boldsymbol{x})$ with $q(f^c | \boldsymbol{y}, \boldsymbol{x})$:

$$
\begin{aligned}
\mathcal{L}_t &\approx \sum_{c \in C} \sum_{t \in T} \mathbb{E}_{q(\boldsymbol{z}_S^c | \boldsymbol{y}_S)} \left[ \mathbb{E}_{q(\boldsymbol{z}_T^c | \boldsymbol{y}_T)} \left[ \mathbb{E}_{q(f^c | \boldsymbol{y}, \boldsymbol{x})} \left[ \log \frac{q(\boldsymbol{z}_t^c | \boldsymbol{y}_t)}{p\left(\boldsymbol{z}_t^c | f^c, \boldsymbol{z}_S^c, \boldsymbol{x}_S, \boldsymbol{x}_t\right)} \right] \right] \right] \\
&= \sum_{c \in C} \sum_{t \in T} \mathbb{E}_{q(\boldsymbol{z}_S^c | \boldsymbol{y}_S)} \left[ \mathbb{E}_{q(f^c | \boldsymbol{y}, \boldsymbol{x})} \left[ \mathrm{KL} \big( q(\boldsymbol{z}_t^c | \boldsymbol{y}_t) \| p\left(\boldsymbol{z}_t^c | f^c, \boldsymbol{z}_S^c, \boldsymbol{x}_S, \boldsymbol{x}_t\right) \big) \right] \right].
\end{aligned}
\tag{17}
$$

In this way, we convert the computation of the log-likelihoods on $q(\boldsymbol{z}_t^c | \boldsymbol{y}_t)$ and $p(\boldsymbol{z}_t^c | f^c, \boldsymbol{z}_S^c, \boldsymbol{x}_S, \boldsymbol{x}_t)$ to the KL divergences between them. Then a Monte Carlo estimator can be used to approximate $\mathcal{L}_t$ where $\tilde{\boldsymbol{z}}_S^c$ is sampled by $\tilde{\boldsymbol{z}}_S^c \sim q(\boldsymbol{z}_S^c | \boldsymbol{y}_S)$ to compute the outer expectation $\mathbb{E}_{q(\boldsymbol{z}_S^c | \boldsymbol{y}_S)}[*]$; and for inner expectation $\mathbb{E}_{q(f^c | \boldsymbol{y}, \boldsymbol{x})}[*]$, because $q(f^c | \boldsymbol{y}, \boldsymbol{x}) = \int q(f^c | \boldsymbol{z}^c, \boldsymbol{x}) q(\boldsymbol{z}^c | \boldsymbol{y}) d\boldsymbol{z}^c$, we are able to first sample $\tilde{\boldsymbol{z}}_T^c \sim q(\boldsymbol{z}_T^c | \boldsymbol{y}_T)$ and obtain $\tilde{f}^c \sim q(f^c | \tilde{\boldsymbol{z}}_S^c, \tilde{\boldsymbol{z}}_T^c, \boldsymbol{x})$. By means of $\tilde{f}^c$ and $\tilde{\boldsymbol{z}}^c$ sampled from the posterior, we have $\mathcal{L}_t \approx \mathrm{KL}(q(\boldsymbol{z}_t^c | \boldsymbol{y}_t) \| p(\boldsymbol{z}_t^c | \tilde{f}^c, \tilde{\boldsymbol{z}}_S^c, \boldsymbol{x}_S, \boldsymbol{x}_t))$.

### B.3.3 CONTEXT REGULARIZER

$$
\mathcal{L}_s = \sum_{c \in C} \sum_{s \in S} \mathbb{E}_{q(\boldsymbol{h}|\boldsymbol{y},\boldsymbol{x})} \left[ \log \frac{q(\boldsymbol{z}_s^c | \boldsymbol{y}_s)}{p\left(\boldsymbol{z}_s^c | \boldsymbol{y}_s\right)} \right] = \sum_{c \in C} \sum_{s \in S} \mathbb{E}_{q(\boldsymbol{h}|\boldsymbol{y},\boldsymbol{x})} \big[ \log 1 \big] = 0.
\tag{18}
$$

### B.3.4 FUNCTION REGULARIZER

$$
\begin{aligned}
\mathcal{L}_f &= \sum_{c \in C} \mathbb{E}_{q(\boldsymbol{h}|\boldsymbol{y},\boldsymbol{x})} \left[ \log \frac{q(f^c|\boldsymbol{z}^c,\boldsymbol{x})}{p\left(f^c|\boldsymbol{z}_S^c,\boldsymbol{x}_S\right)} \right] \\
&= \sum_{c \in C} \mathbb{E}_{q(\boldsymbol{z}_S^c|\boldsymbol{y}_S)} \left[ \mathbb{E}_{q(\boldsymbol{z}_T^c|\boldsymbol{y}_T)} \left[ \mathbb{E}_{q(f^c|\boldsymbol{z}^c,\boldsymbol{x})} \left[ \log \frac{q(f^c|\boldsymbol{z}^c,\boldsymbol{x})}{p\left(f^c|\boldsymbol{z}_S^c,\boldsymbol{x}_S\right)} \right] \right] \right] \\
&= \sum_{c \in C} \mathbb{E}_{q(\boldsymbol{z}^c|\boldsymbol{y})} \left[ \mathrm{KL}\big(q(f^c|\boldsymbol{z}^c,\boldsymbol{x}) \| p\left(f^c|\boldsymbol{z}_S^c,\boldsymbol{x}_S\right)\big) \right].
\end{aligned}
\tag{19}
$$

To estimate $\mathcal{L}_f$ in the same way as the target regularizer, $\tilde{\boldsymbol{z}}^c$ is sampled from $q(\boldsymbol{z}^c|\boldsymbol{y})$ and we have $\mathcal{L}_f \approx \mathrm{KL}(q(f^c|\tilde{\boldsymbol{z}}^c,\boldsymbol{x}) \| p\left(f^c|\tilde{\boldsymbol{z}}_S^c,\boldsymbol{x}_S\right))$.

### B.4 TOTAL CORRELATION

Let $\mathcal{I} = \{\boldsymbol{y}_k\}_{k=1}^K$ denote the set of all sample images in the dataset, and $\tilde{\boldsymbol{z}}_k \sim q(\boldsymbol{z}_k|\boldsymbol{y}_k)$ are the latent representations of concepts for the $k$th image. To apply the Total Correlation (TC) (Chen et al., 2018) in CLAP, we decompose the aggregated posterior $\bar{q}(\boldsymbol{z}) = \sum_{k=1}^K q(\boldsymbol{z}_k|\boldsymbol{y}_k)p(\boldsymbol{y}_k)$ according to the concepts, that is

$$
\mathcal{R}_{TC} = \mathrm{KL}\left( \bar{q}(\boldsymbol{z}) \Big\| \prod_{c \in C} \bar{q}(\boldsymbol{z}^c) \right) = \mathbb{E}_{\bar{q}(\boldsymbol{z})}\big[ \log \bar{q}(\boldsymbol{z}) \big] - \sum_{c \in C} \mathbb{E}_{\bar{q}(\boldsymbol{z})}\big[ \log \bar{q}(\boldsymbol{z}^c) \big].
\tag{20}
$$

We adopt Minibatch Weighted Sampling (Chen et al., 2018) to approximate $\mathcal{R}_{TC}$ on the minibatch $\{\boldsymbol{y}_m\}_{m=1}^M \subseteq \mathcal{I}$ of the dataset and the corresponding representations of concepts $\{\tilde{\boldsymbol{z}}_m\}_{m=1}^M$:

$$
\begin{aligned}
\mathbb{E}_{\bar{q}(\boldsymbol{z})}\big[ \log \bar{q}(\boldsymbol{z}) \big] &\approx \frac{1}{M} \sum_{i=1}^M \left[ \log \sum_{j=1}^M q\left(\tilde{\boldsymbol{z}}_i|\boldsymbol{y}_j\right) - \log(KM) \right], \\
\mathbb{E}_{\bar{q}(\boldsymbol{z})}\big[ \log \bar{q}(\boldsymbol{z}^c) \big] &\approx \frac{1}{M} \sum_{i=1}^M \left[ \log \sum_{j=1}^M q\left(\tilde{\boldsymbol{z}}_i^c|\boldsymbol{y}_j\right) - \log(KM) \right].
\end{aligned}
\tag{21}
$$

## C PRELIMINARIES

**Generative Query Network (GQN)** GQN (Eslami et al., 2018) regards the mappings from camera poses to scene images as random functions. GQN adopts deterministic neural scene representations $\boldsymbol{r}$ and latent variables $\boldsymbol{z}$ to capture the configuration and stochasticity of scenes in the conditional probability $p(\boldsymbol{y}_T|\boldsymbol{x}_T,\boldsymbol{y}_C,\boldsymbol{x}_C) = \int p(\boldsymbol{y}_T,\boldsymbol{z}|\boldsymbol{x}_T,\boldsymbol{r})d\boldsymbol{z}$. The representation network $\mathcal{T}_a$ extracts the representation for each context point and summarizes them as the global scene representation $\boldsymbol{r} = \sum_{c \in C} \mathcal{T}_a(\boldsymbol{y}_c,\boldsymbol{x}_c)$. The generation network $\mathcal{T}_p$ predicts the mean of target scene images via $\boldsymbol{\mu} = \mathcal{T}_p(\boldsymbol{r},\boldsymbol{z},\boldsymbol{x}_t)$, and the target scene images are sampled from $\mathcal{N}(\boldsymbol{y}_t|\boldsymbol{\mu},\sigma^2\boldsymbol{I})$.

**Gaussian Process (GP)** GP (we only discuss noise-free GP here) models the probability distribution $p(\boldsymbol{y}|\boldsymbol{x})$ as the multivariate Gaussian $\boldsymbol{y} \sim \mathcal{N}(\boldsymbol{0},\boldsymbol{K})$ where $\boldsymbol{K}_{ij}$ is computed through the kernel function $\kappa(\boldsymbol{x}_i,\boldsymbol{x}_j)$. In this case, the probability $p(\boldsymbol{y}_T|\boldsymbol{x}_T,\boldsymbol{y}_C,\boldsymbol{x}_C)$ is a multivariate Gaussian, and the parameters have closed-form solutions. Kernel functions are the key to constructing different types of random functions. The models can use a basic kernel like the RBF kernel, combine different kernels to develop complex kernels, or learn the function space for different data adaptively through neural networks (Wilson et al., 2016).

## D DATASETS AND EXPERIMENTAL SETUP

### D.1 DETAILS OF DATASETS

In this paper, three types of datasets are adopted: BoBa, CRPM, and MPI3D. Figure 8 displays two samples for each instance of datasets, and Table 2 describes the train, validation, and test configurations of all instances.

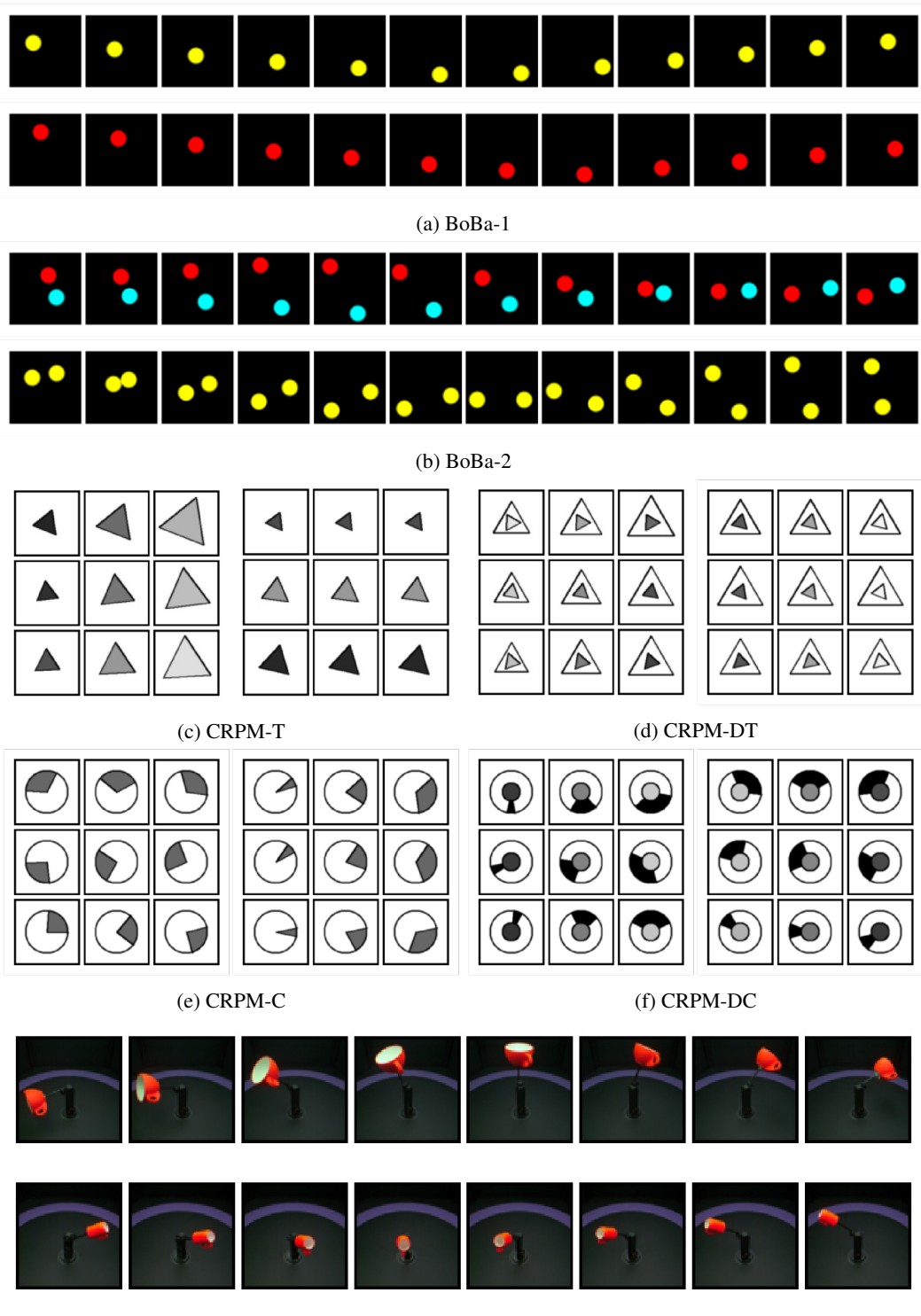

Figure 8: Examples from different instances of datasets. BoBa includes two instances (a) BoBa-1 and (b) BoBa-2; CRPM includes four instances (c) CRPM-T, (d) CRPM-DT, (e) CRPM-C, and (f) CRPM-DC; (g) we only use one instance of MPI3D.

Table 2: The detail of BoBa, CRPM, and MPI3D. Row 1: dataset names. Row 2: splits of datasets where Test-$k$ denotes that there are $k$ target images in one sample. Row 3: the number of samples in each split. Row 4: the number of images in each sample. Row 5: the number of target images in a sample. Row 6: the size of images denoted as Channel $\times$ Height $\times$ Width.

| Dataset | BoBa-1 | | | | BoBa-2 | | | |
|---|---|---|---|---|---|---|---|---|
| Split | Train | Valid | Test-4 | Test-6 | Train | Valid | Test-4 | Test-6 |
| Samples | 10000 | 1000 | 2000 | 2000 | 10000 | 1000 | 2000 | 2000 |
| Images | 12 | | | | 12 | | | |
| Targets | $1 \sim 4$ | $1 \sim 4$ | 4 | 6 | $1 \sim 4$ | $1 \sim 4$ | 4 | 6 |
| Image Size | $3 \times 64 \times 64$ | | | | $3 \times 64 \times 64$ | | | |
| Dataset | CRPM-T | | | | CRPM-C | | | |
| Split | Train | Valid | Test-2 | Test-3 | Train | Valid | Test-2 | Test-3 |
| Samples | 10000 | 1000 | 2000 | 2000 | 10000 | 1000 | 2000 | 2000 |
| Images | 9 | | | | 9 | | | |
| Targets | $1 \sim 2$ | $1 \sim 2$ | 2 | 3 | $1 \sim 2$ | $1 \sim 2$ | 2 | 3 |
| Image Size | $1 \times 64 \times 64$ | | | | $1 \times 64 \times 64$ | | | |
| Dataset | CRPM-DT | | | | CRPM-DC | | | |
| Split | Train | Valid | Test-2 | Test-3 | Train | Valid | Test-2 | Test-3 |
| Samples | 10000 | 1000 | 2000 | 2000 | 10000 | 1000 | 2000 | 2000 |
| Images | 9 | | | | 9 | | | |
| Targets | $1 \sim 2$ | $1 \sim 2$ | 2 | 3 | $1 \sim 2$ | $1 \sim 2$ | 2 | 3 |
| Image Size | $1 \times 64 \times 64$ | | | | $1 \times 64 \times 64$ | | | |

| Dataset | MPI3D | | | | | |
|---|---|---|---|---|---|---|
| Split | Train | Valid | Test-4 | Test-6 | Test-10 | Test-20 |
| Samples | 16127 | 2305 | 4608 | 4608 | 4608 | 4608 |
| Images | 8 | | | | 40 | |
| Targets | $1 \sim 4$ | $1 \sim 4$ | 4 | 6 | 10 | 20 |
| Image Size | $3 \times 64 \times 64$ | | | | | |

Table 3: A detailed description of $x_n$ on datasets.

| Dataset | Description |
|---|---|
| CRPM | 2D index of the image in matrix |
| BoBa | timestamps of the frame |
| MPI3D | azimuths or altitudes of the center object |

**Bouncing Ball (BoBa)**     BoBa contains videos of bouncing balls. Depending on the number of balls in the scene, the dataset provides two instances BoBa-1 and BoBa-2. In the videos of Figure 8a and 8b, the motion of balls follows the law of physical collisions; and the appearance of balls (color, size, amount, etc.) is constant over time. The models need to capture the probability space on functions $f : \mathbb{R} \mapsto \mathbb{R}^{3 \times 64 \times 64}$ that map timestamps to video frames. Referring to Table 2, we provide 10,000 videos of bouncing balls for training, 1,000 for validation and hyperparameter selection, and 2,000 to test the intuitive physics of the models. In the training and validation phase, we randomly select 1 to 4 frames from video frames as the target images to predict; in the testing phase, we provide two configurations (referred to as Test-4 and Test-6) to evaluate the model's performance when there are 4 or 6 target images in videos.

Table 4: The hyperparameters of CLAP-NP.

| Dataset | $|C|$ | $\beta_t$ | $\beta_f$ | $\beta_{TC}$ | $\sigma_z$ | $d_g$ |
|---------|-------|-----------|-----------|--------------|------------|-------|
| BoBa-1  | 3 | 100 | 100 | 1000  | 0.03 | 32 |
| BoBa-2  | 6 | 400 | 100 | 1500  | 0.03 | 64 |
| CRPM-T  | 3 | 400 | 100 | 5000  | 0.1  | 64 |
| CRPM-DT | 3 | 200 | 50  | 5000  | 0.1  | 64 |
| CRPM-C  | 2 | 100 | 100 | 5000  | 0.1  | 64 |
| CRPM-DC | 3 | 300 | 300 | 10000 | 0.1  | 64 |
| MPI3D   | 7 | 50  | 50  | 150   | 0.03 | 64 |

**Continuous Raven's Progressive Matrix (CRPM)** CRPM consists of $3 \times 3$ image matrices where the images contain one or two centered triangles or circles. CRPM provides four instances with different image types: CRPM-T, CRPM-DT, CRPM-C, and CRPM-DC. Images in a matrix follow attribute-specific changing rules (e.g., rules in the size, grayscale, and rotation of the triangle). The first sample in Figure 8c shows that, in the row of the matrix, the sizes of triangles increase progressively, the grayscales change progressively from dark to light, and the rotations keep constant. To predict the missing target images in the matrix, the models need to learn the probability space on functions $f : \{-1, 0, 1\}^2 \mapsto \mathbb{R}^{64 \times 64}$ that map 2D grid coordinates to grid images. Table 2 shows that we provide 10,000 image matrices for training, 1,000 for tuning the model hyperparameters, and 2,000 to test the abstract visual reasoning ability of models. In the training and validation process, we randomly select 1 or 2 images as target images; in the testing process, we use Test-2 and Test-3 to evaluate the performance of the models to predict 2 or 3 target images.

**MPI3D** MPI3D (Gondal et al., 2019) dataset contains a series of single-object scenes, each with 40 different scene images. There are two underlying rules to change the object in a scene: change the altitude of the object (sample 1 in Figure 8g); change the azimuth of the object (sample 2 in Figure 8g). The other attributes (e.g., object color, camera height, etc.) do not change within scene images. It is essential for the models to grasp different changing patterns for target image prediction, and the key is to describe the distribution over functions $f : \mathbb{R} \mapsto \mathbb{R}^{3 \times 64 \times 64}$ that map object altitudes or azimuths to scene images. As Table 2 shows, we provide 16127 scenes for training, 2305 for tuning the hyperparameters, and 4608 for testing. In terms of training and validation, we first randomly select 8 images from 40 scene images to represent the scene, and then randomly select 1 to 4 images from the 8 images again as the target images, leaving the remaining images as the context. For testing, we supply Test-4, Test-6, Test-10, and Test-20, where Test-4 and Test-6 evaluate the performance of the models to predict 4 or 6 target images out of 8 images; Test-10 and Test-20 evaluate the performance to predict 10 or 20 target images out of 40 images. One can fetch MPI3D from the repository [2] under the Creative Commons Attribution 4.0 International License.

## D.2 MODEL ARCHITECTURE AND HYPERPARAMETERS

**CLAP-NP** In this subsection, we will first describe the architecture of the encoder, decoder, concept-specific function parsers, and concept-specific target predictors in CLAP-NP. Then we will list hyperparameters in the training phase.

- **Encoder**: for all datasets, CLAP-NP uses the same convolutional blocks to downsample high-dimensional images into the representations of concepts, which can be described as

  - $4 \times 4$ Conv, stride 2, padding 1, 32 BatchNorm, ReLU
  - $4 \times 4$ Conv, stride 2, padding 1, 32 BatchNorm, ReLU
  - $4 \times 4$ Conv, stride 2, padding 1, 64 BatchNorm, ReLU
  - $4 \times 4$ Conv, stride 2, padding 1, 128 BatchNorm, ReLU
  - $4 \times 4$ Conv, 256 BatchNorm, ReLU
  - $1 \times 1$ Conv, 512
  - ReshapeBlock, 512

---

[2] https://github.com/rr-learning/disentanglement_dataset

- Fully Connected, $|C|$

The ReshapeBlock flattens the output tensor of size $512 \times 1 \times 1$ from the last convolutional layer to the vector of size $512$. The output of the encoder is the mean of $|C|$ concepts.

- **Decoder**: CLAP-NP uses several deconvolutional layers to upsample the representations of concepts to images, which is

    - ReshapeBlock, $|C| \times 1 \times 1$
    - $1 \times 1$ Deconv, 128 BatchNorm, ReLU
    - $4 \times 4$ Deconv, 64 BatchNorm2d, ReLU
    - $4 \times 4$ Deconv, stride 2, padding 1, 32 BatchNorm, ReLU
    - $4 \times 4$ Deconv, stride 2, padding 1, 32 BatchNorm, ReLU
    - $4 \times 4$ Deconv, stride 2, padding 1, 32 BatchNorm, ReLU
    - $4 \times 4$ Deconv, stride 2, padding 1, $N_{\text{Channel}}$ Sigmoid

    where $N_{\text{Channel}}$ is the number of image channels and Sigmoid is the activation function used to generate pixel values ranging in $(0, 1)$.

- **Function Parser**: CLAP-NP provides identical but independent function parsers for concepts. Each function parser consists of the network $\mathcal{T}_s$ to encode the representations of concepts, $\mathcal{T}_i$ to encode the inputs of latent random functions, $\mathcal{T}_a$ to aggregate context information, and $\mathcal{T}_f$ to estimate the distribution of the global latent variable. The architecture of $\mathcal{T}_s$ and $\mathcal{T}_i$ is

    - Fully Connected, 32 ReLU
    - Fully Connected, 16

    By concatenating the representations of concepts and the function inputs, we get the context points in the latent random function. The architecture of $\mathcal{T}_a$ is

    - Fully Connected, 256 ReLU
    - Fully Connected, 256 ReLU
    - Fully Connected, 128

    Finally, the architecture of $\mathcal{T}_f$ is

    - Fully Connected, 256 ReLU
    - Fully Connected, $2d_g$

    where $d_g$ is the size of the global latent variable $\boldsymbol{g}^c$. The outputs of $\mathcal{T}_f$ consist of the mean and standard deviation of $\boldsymbol{g}^c$.

- **Target Predictor**: the concept-specific target predictor $\mathcal{T}_p$ maps $\boldsymbol{g}^c$ and the encoded function inputs into the target concept, whose architecture is

    - Fully Connected, 256 ReLU
    - Fully Connected, 256 ReLU
    - Fully Connected, 1

To generate more complex scene images in MPI3D, CLAP-NP uses a deeper decoder with hidden sizes [512, 256, 128, 64, 32]. And we use $\mathcal{T}_f$ with hidden sizes [128, 128] and output size 64, $\mathcal{T}_f$ with hidden size [64], and $\mathcal{T}_p$ with hidden sizes [128, 128]. Hyperparameters for CLAP-NP on different datasets are shown in Table 4. In all datasets, we set the learning rate as $3 \times 10^{-4}$, batch size as 512, and $\sigma_y = 0.1$. For MPI3D, we anneal $\beta_t$ and $\beta_f$ in the first 400 epochs. After each training epoch, we use the validation set to compute the evidence lower bound (ELBO) of the current model and save the model with the largest ELBO as the trained model. For all datasets, CLAP-NP uses the Adam (Kingma & Ba, 2015) optimizer to update parameters.

**Neural Process (NP)**    To deal with high-dimensional images in NP (Garnelo et al., 2018b), we apply the encoder and decoder of CLAP-NP in it to convert high-dimensional images into low-dimensional representations. The aggregator $\mathcal{T}_a$ extract context information from the representations and function inputs. Then the function parser $\mathcal{T}_f$ converts the context representation into the mean and standard deviation of the global latent variable. Finally, the decoder generates target images from the target function inputs and the global latent variable. The architectures of $\mathcal{T}_a$ and $\mathcal{T}_f$ are

- **Aggregator**

Table 5: The hyperparameters of NP.

| Dataset | $d_r$ | $d_g$ |
|---------|-------|-------|
| BoBa-1  | 3     | 1024  |
| BoBa-2  | 6     | 1024  |
| CRPM-T  | 3     | 1024  |
| CRPM-DT | 3     | 1024  |
| CRPM-C  | 2     | 512   |
| CRPM-DC | 3     | 512   |
| MPI3D   | 7     | 1024  |

Table 6: The learning rate and training epoch of GP.

| Dataset | learning rate | epoch |
|---------|---------------|-------|
| BoBa-1  | 0.05          | 200   |
| BoBa-2  | 0.01          | 200   |
| CRPM-T  | 0.05          | 200   |
| CRPM-DT | 0.05          | 100   |
| CRPM-C  | 0.01          | 400   |
| CRPM-DC | 0.05          | 100   |
| MPI3D   | 0.05          | 10    |

- – Fully Connected, 512 ReLU

- – Fully Connected, 512 ReLU

- – Fully Connected, 512

- **Function Parser**

  - – Fully Connected, 512 ReLU

  - – Fully Connected, 512 ReLU

  - – Fully Connected, $2d_g$

NP's hyperparameters on different datasets are shown in Table 5 where $d_r$ is the representation size of the encoder. In all datasets, NP adopts the learning rate $1 \times 10^{-4}$ batch size 512, and $\sigma_y = 0.1$. The parameters of NP are updated with the Adam optimizer (Kingma & Ba, 2015).

**Generative Query Network (GQN)**    To implement GQN (Eslami et al., 2018), we use a PyTorch implementation from the repository [3] with the following changes to the default configuration to control the computational resource: set learning rate to 0.0005, batch size to 256, the representation type to pool, the number of iterations to 4, and share the ConvLSTM core among iterations.

**Gaussian Process (GP)**    We use a pretrained autoencoder to reduce the dimension of images and let GP capture the changing patterns in the low-dimensional space. We adopt the same encoder and decoder as in CLAP-NP and NP for a fair comparison. The mean function of GP is set to a constant function. After comparing the RBF kernel, periodic kernel, and deep kernel, we chose the deep kernel as the kernel function that $\kappa(\boldsymbol{x}_i, \boldsymbol{x}_j) = \sigma^2 \exp(-\|\mathcal{T}_k(\boldsymbol{x}_i) - \mathcal{T}_k(\boldsymbol{x}_j)\|/2\ell^2)$ (Wilson et al., 2016). Neural network $\mathcal{T}_k$ is a multilayer perceptron with hidden sizes [1000, 1000, 500, 50], output size 2, and ReLU activation functions, which is the same as in DKL (Wilson et al., 2016). The hyperparameters of GP are tuned on each sample through the log-likelihood of context points and the RMSprop optimizer. We give the learning rate and epoch to adjust hyperparameters in Table 6 and use multi-task Gaussian Process prediction (Bonilla et al., 2007) to model the correlations between dimensions.

---

[3]https://github.com/iShohei220/torch-gqn

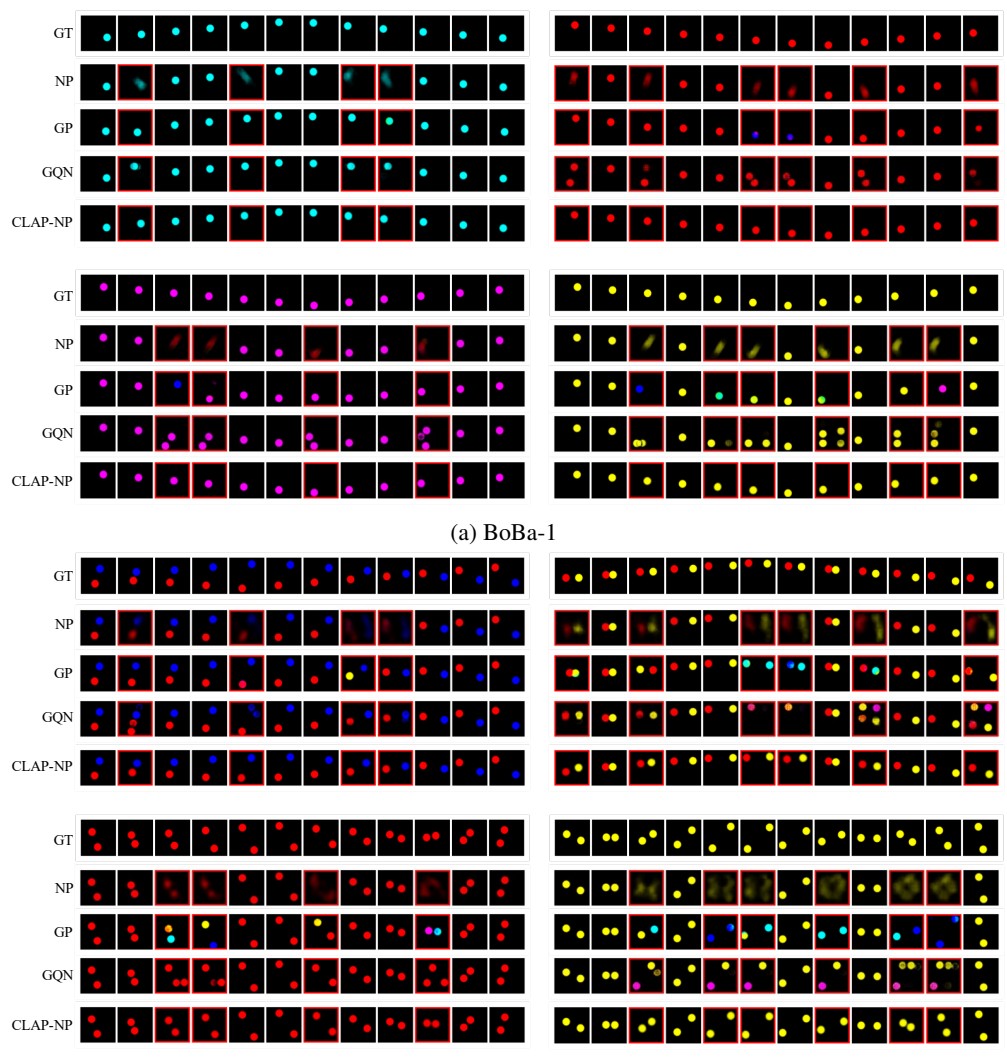

(a) BoBa-1

(b) BoBa-2

Figure 9: Target prediction on BoBa. The images with red boxes are predictions from the models.

## D.3 COMPUTATIONAL RESOURCE

We train our model and baselines on the server with Intel(R) Xeon(R) Gold 6133 CPUs, 24GB NVIDIA GeForce RTX 3090 GPUs, 512GB RAM, and Ubuntu 18.04 OS. All models are implemented with the PyTorch (Paszke et al., 2019) framework.

## E ADDITIONAL EXPERIMENTAL RESULTS

### E.1 INTUITIVE PHYSICS

In Figure 9 we display additional prediction results on BoBa-1 and BoBa-2. For each instance, we provide samples using the configurations Test-4 (left) or Test-6 (right), where CLAP-NP outperforms other baselines. NP generates blurred target images on both BoBa-1 and BoBa-2 datasets. It indicates that NP has difficulty modeling changing patterns on bouncing balls. GQN can produce clear images but may generate non-existent balls and lose existing balls. In the first sample of BoBa-1, the predictions of GQN deviate from the ground truths significantly in the position of balls. CLAP-NP performs well in modeling scene consistency and predicting motion trajectory.

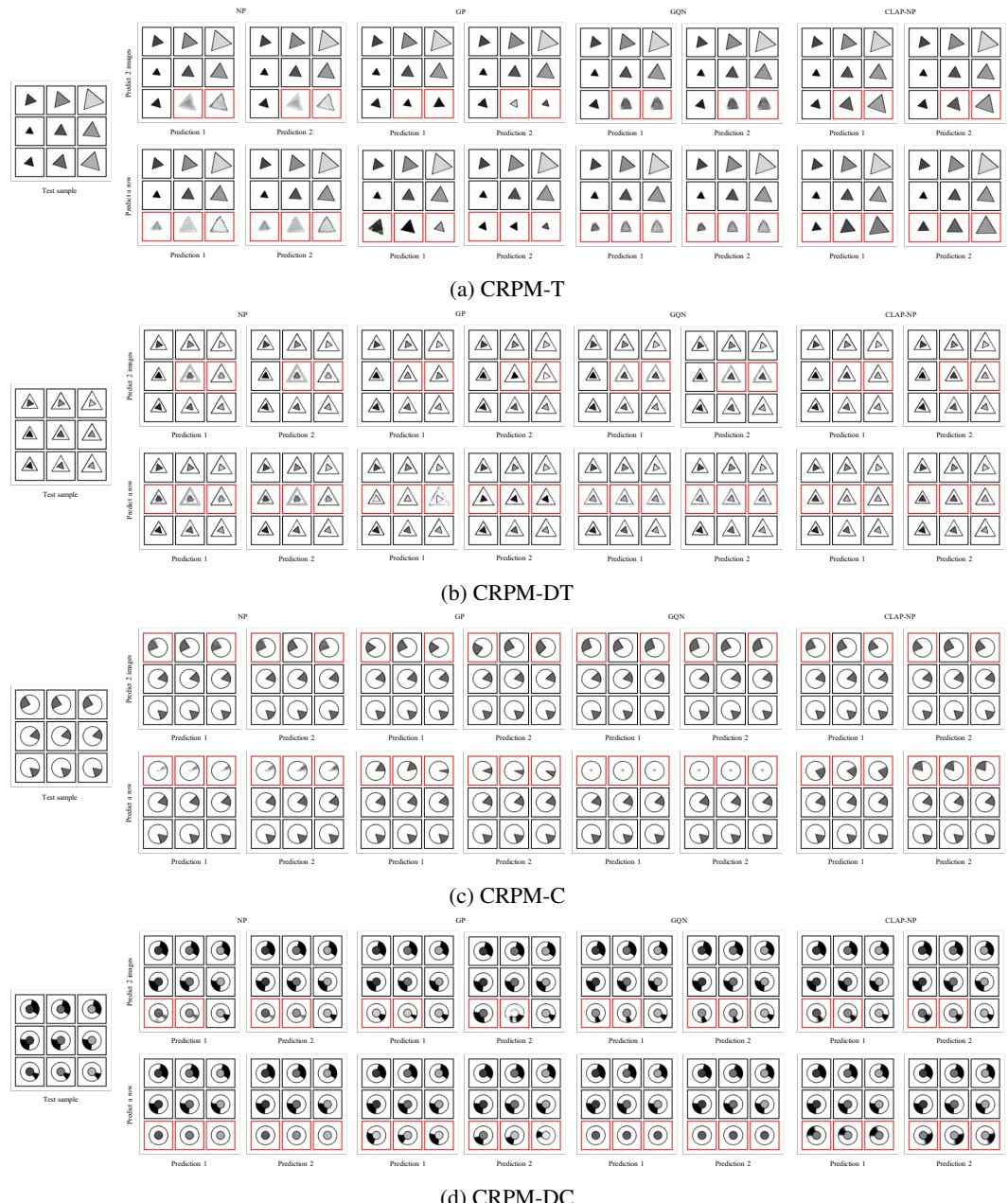

(a) CRPM-T

(b) CRPM-DT

(c) CRPM-C

(d) CRPM-DC

Figure 10: Target predictions on CRPM. For each sample, we show two predictions of the models to test the understanding of attribute-specific rules.

## E.2 ABSTRACT VISUAL REASONING

With the additional experimental results in Figure 10, we can intuitively see that CLAP-NP has a better understanding of the changing rules than baselines. For the second sample of each dataset, a row of images is removed to interpret the abstract visual reasoning ability of the model. As samples in Figure 10 shows, when we remove a row from the matrix, the answer is not unique: the predictions are correct as long as their sizes increase progressively and their colors and rotations keep constant. CLAP-NP represents such randomness in predictions by means of the probabilistic generative process, making it possible to generate different correct answers. In terms of the prediction quality, although generating blurred images, NP has the basic reasoning ability about the progressive rule of

Table 7: MSE scores on CRPM-T and CRPM-C.

| Model | CRPM-T ($\eta = 1{\sim}2/9$) | | CRPM-C ($\eta = 1{\sim}2/9$) | |
| --- | --- | --- | --- | --- |
| | $\eta = 2/9$ | $\eta = 3/9$ | $\eta = 2/9$ | $\eta = 3/9$ |
| NP | $125.8 \pm 4.9$ | $204.6 \pm 5.9$ | $112.8 \pm 6.9$ | $186.8 \pm 12.6$ |
| GP | $416.4 \pm 44.2$ | $611.0 \pm 44.9$ | $286.0 \pm 26.5$ | $432.1 \pm 27.4$ |
| GQN | $235.2 \pm 4.7$ | $493.4 \pm 7.2$ | $246.5 \pm 5.2$ | $484.6 \pm 8.2$ |
| CLAP-NP | $\mathbf{41.3 \pm 1.8}$ | $\mathbf{89.1 \pm 13.7}$ | $\mathbf{52.4 \pm 11.3}$ | $\mathbf{136.6 \pm 15.8}$ |

Figure 11: SA scores on CRPM-T and CRPM-C.

the outer triangle size in Figure 10b. GQN generates clear images, however, the generated images probably deviate from the underlying changing rules on matrices. Table 7 and Table 11 illustrate the outperforming results of CLAP-NP in abstract visual reasoning by quantitative experiments.

### E.3  SCENE REPRESENTATION

Figure 12 shows the prediction results within 8 scene images. Both GQN and CLAP-NP generate clear prediction results when the number of target images is 4; if the number of target images is increased to 6, the prediction accuracy of GQN has a significant decline, while CLAP-NP maintains the accuracy. NP generates ambiguous results in all experiments. Figure 13 shows a more complicated situation: we provide 40 scene images and set 20 of them as target images. In this case, GQN and NP can hardly generate clear foreground objects; while CLAP-NP produces relatively accurate predictions with only a little decrease in generative quality. This experiment aims to test the generalization of the laws learned by the models, and the results above illustrate that NP can hardly represent scenes with functions, GQN has difficulty generalizing the scene representation ability to different configurations, and the laws learned by CLAP-NP can be generalized to novel scenes with the compositional modeling of laws.

### E.4  CONCEPT DECOMPOSITION

Concept decomposition is the foundation for CLAP to understand concept-specific laws. In this experiment, we traverse each concept in the latent space and visualize the concepts through the decoder to illustrate the LRFs. First of all, we decompose a batch of images into concepts by the encoder to estimate the range of concepts in the latent space. To traverse one concept, we fix the other concepts and linearly interpolate it from the minimum value to the maximum value to generate a sequence of interpolation results, which are decoded into images for visualization. Each row of Figure 14 represents the traversal results of one concept. In BoBa-1, CLAP-NP learns LRFs on concepts of color, horizontal position, and vertical position in an unsupervised manner. This is similar to the way we understand the motion of balls: the color keeps constant over time; the horizontal and vertical positions conform to the physical laws in their respective directions. For CRPM, CLAP-NP correctly parses images into concepts that correspond to the attribute-specific rules in matrices. Concept decomposition in real environments is a challenge for models. For MPI3D, CLAP-NP parses the LRFs on the object appearance (Dimension 1, 5, 6, 7) and other static scene attributes (Dimension 2, 3, 4).

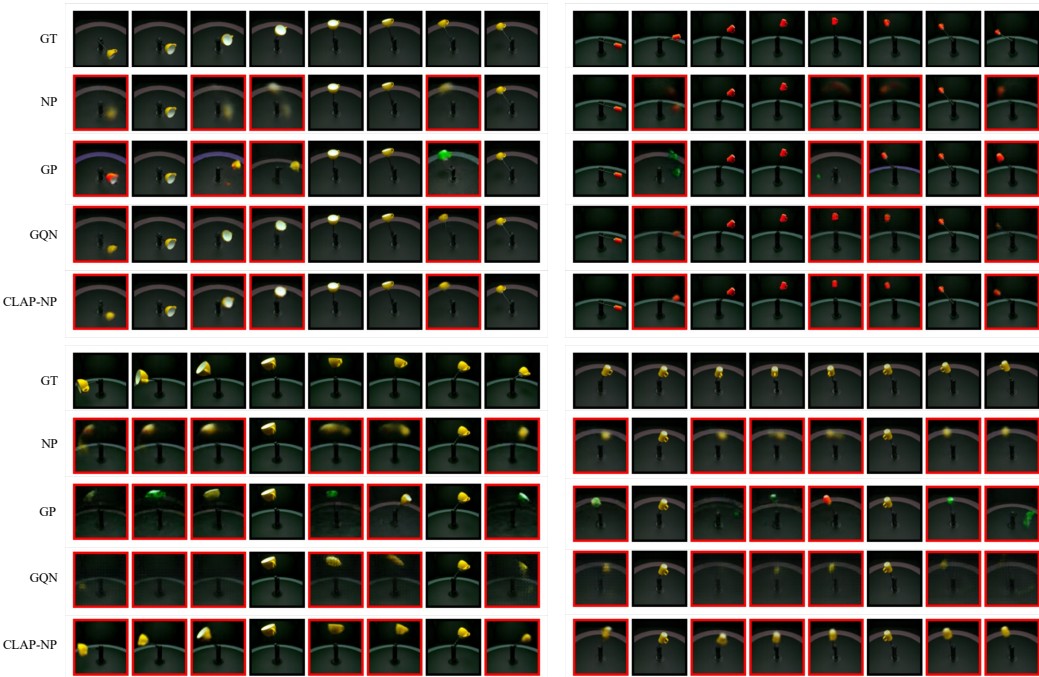

Figure 12: Target prediction with 8 scene images on MPI3D

## E.5 Latent Random Function Composition

By means of concept decomposition, CLAP-NP can parse the LRF on each concept to represent concept-specific laws, by which the law of a sample can be modified in terms of concepts. Figure 15 and 16 display the way to manipulate the laws. We compose the concept-specific LRFs of existing samples to generate samples that have novel changing patterns. For example, in the third sample of Figure 15a, we generate a sample with the progressive law of size, constant law of rotation, and constant law of color from three existing samples. The results on four instances of CRPM indicate that CLAP-NP understands the changing patterns on matrices in an interpretable and manipulable way, which embodies the compositionality of laws.

## E.6 Latent Random Function Exchange

Another way to manipulate laws is to exchange LRFs of some concepts between samples. In BoBa, we swap the law of appearance or object motion between samples, and the results are shown in Figure 17. According to the concept decomposition results in Figure 14, we exchange the LRFs between samples on Dimension 1 (Dimensions 2 and 3) to modify the motion of balls (the law of appearance) in BoBa-1. And we exchange the LRFs between samples on Dimensions 1 and 6 (Dimensions 2, 3, 4, and 5) to modify the motion of balls (the law of appearance) in BoBa-2. To exchange laws in MPI3D, as shown in Figure 18, we swap LRFs between samples on Dimensions 1, 5, 6, and 7 to change the law of the object motion. By modifying laws on example 1 of Figure 18, the changing pattern of the first sample becomes the horizontal rotation, and the second sample inherits the law of vertical rotation from the first sample, which illustrates CLAP-NP's ability to generate new samples. The experiments on law exchange evidence CLAP-NP's interpretability and manipulability from another perspective.

## E.7 Generalization on Unseen Concepts

We extend BoBa-2 to generate four datasets with novel laws: **Novel-colors** (BoBa-2-NC), **Novel-shape** (BoBa-2-NS), **Without-ball-collisions** (BoBa-2-WBO), and **With-gravity** (BoBa-2-WG). We draw balls on BoBa-2-NC with unseen colors and replace the balls on BoBa-2-NS with squares without changing the laws of motion. For BoBa-2-WBO, we disable collisions between balls and re-

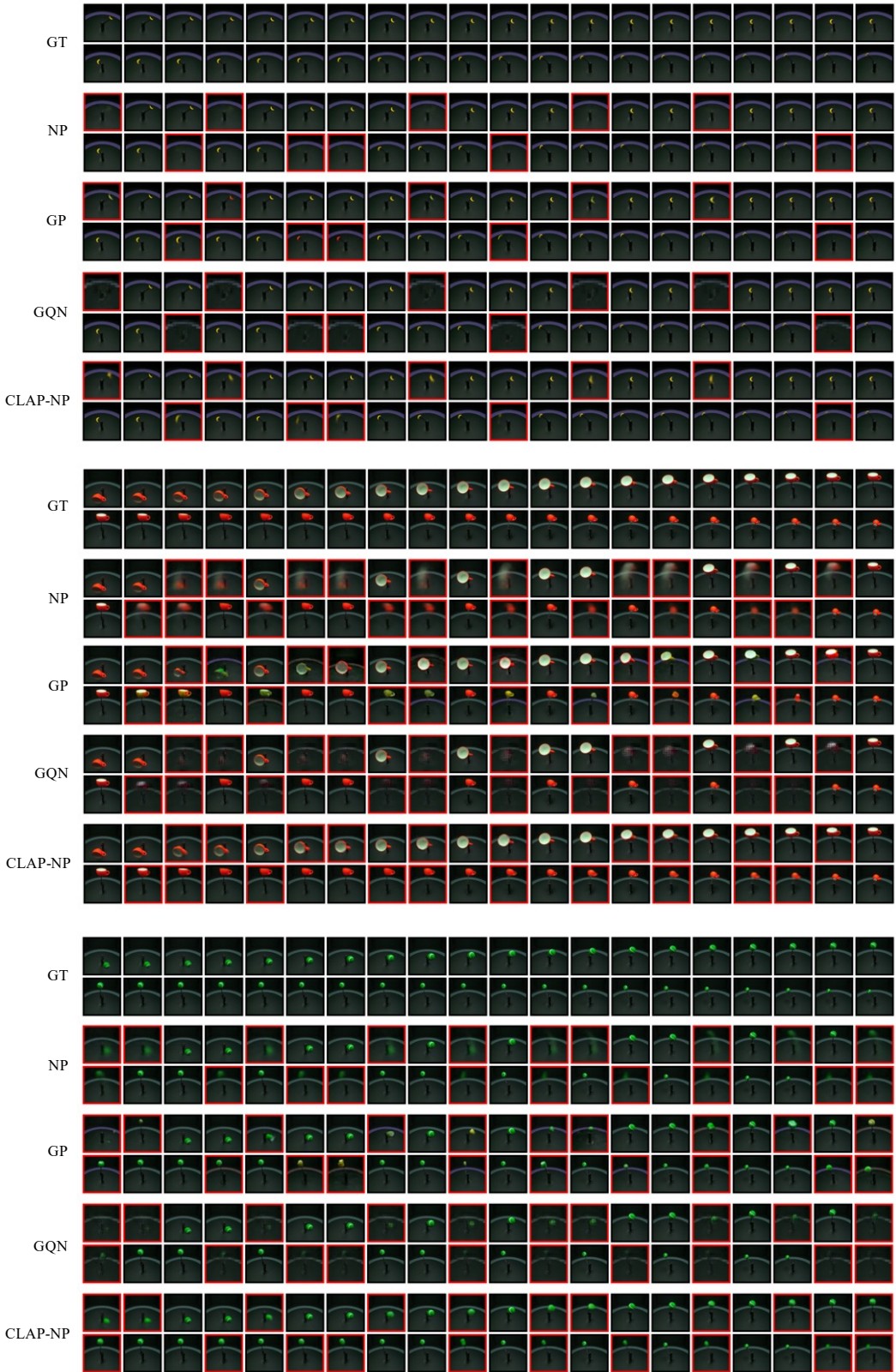

Figure 13: Target prediction with 40 scene images on MPI3D

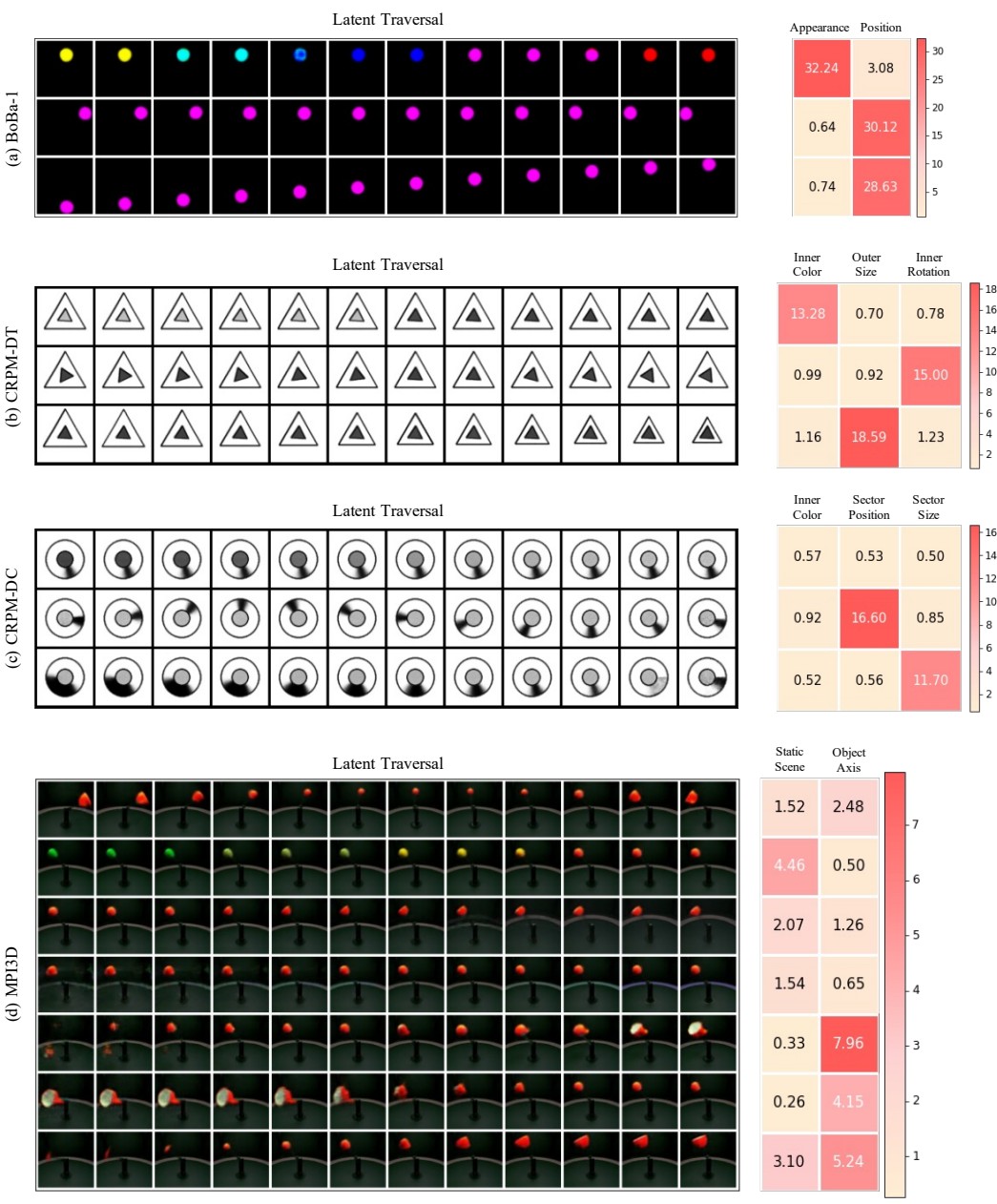

Figure 14: Latent traversal results and variance declines on BoBa-1, CRPM-DT, CRPM-DC, and MPI3D.

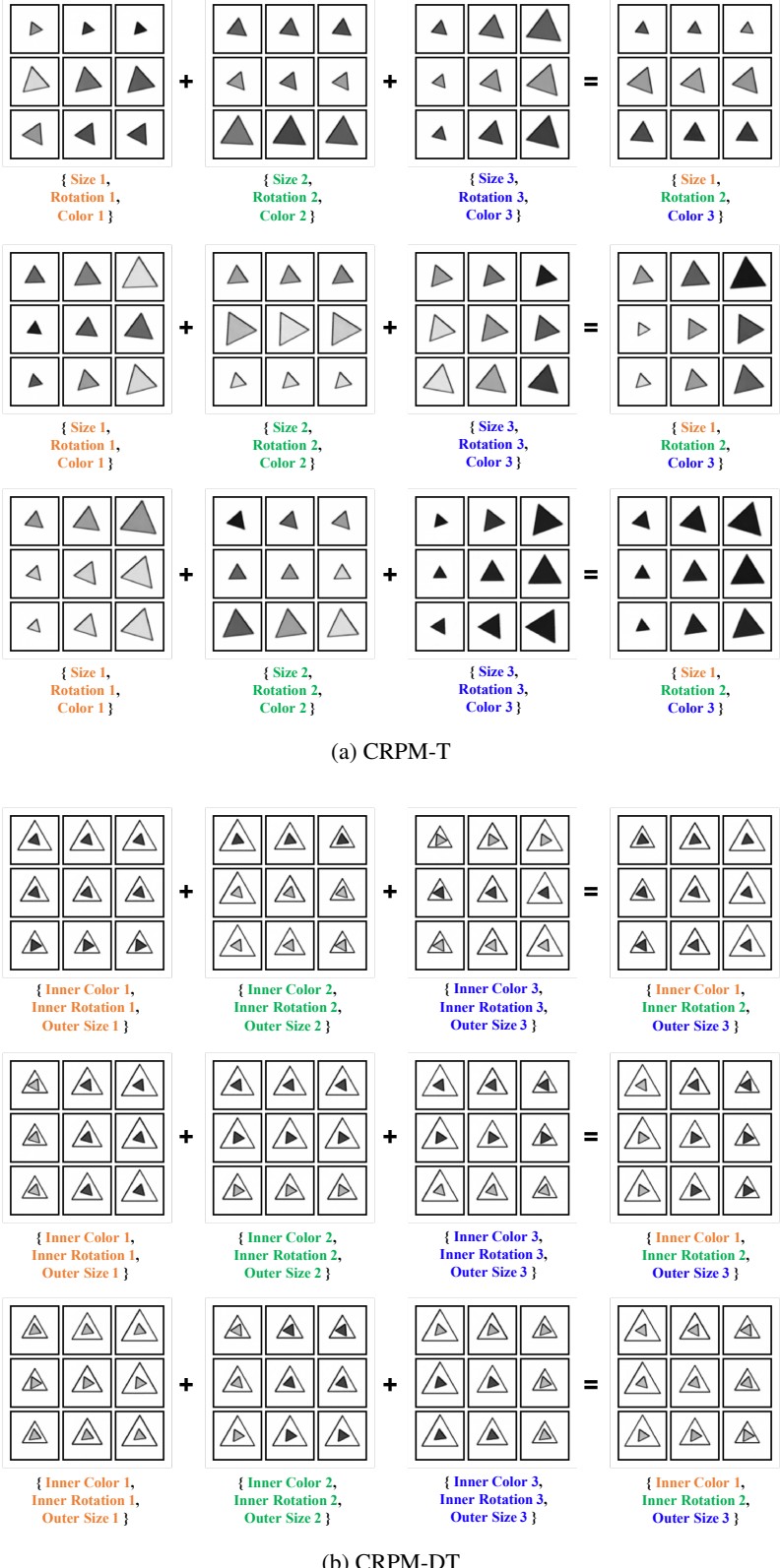

Figure 15: Law Composition on CRPM-T and CRPM-DT.

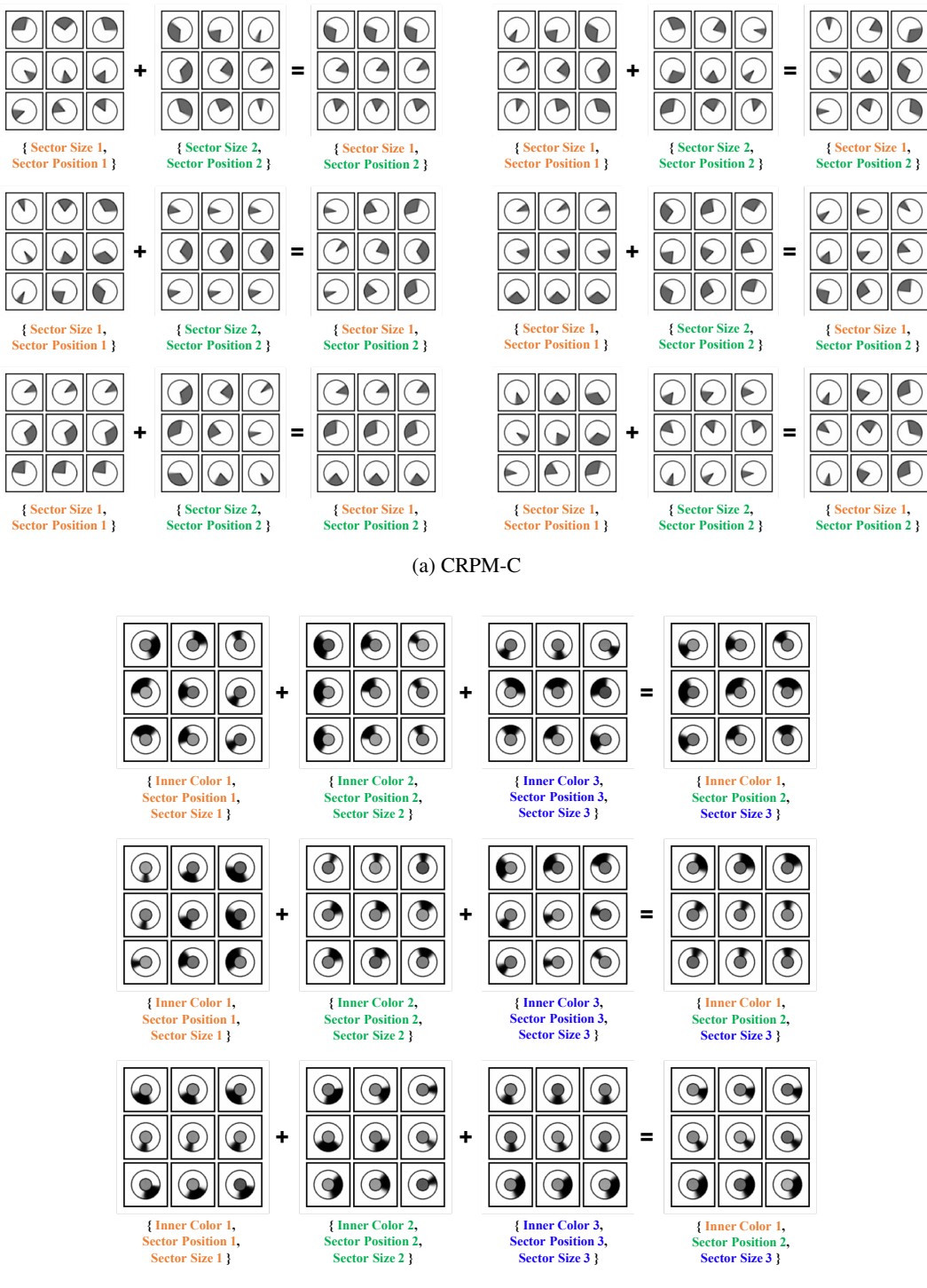

Figure 16: Law Composition on CRPM-C and CRPM-DC.

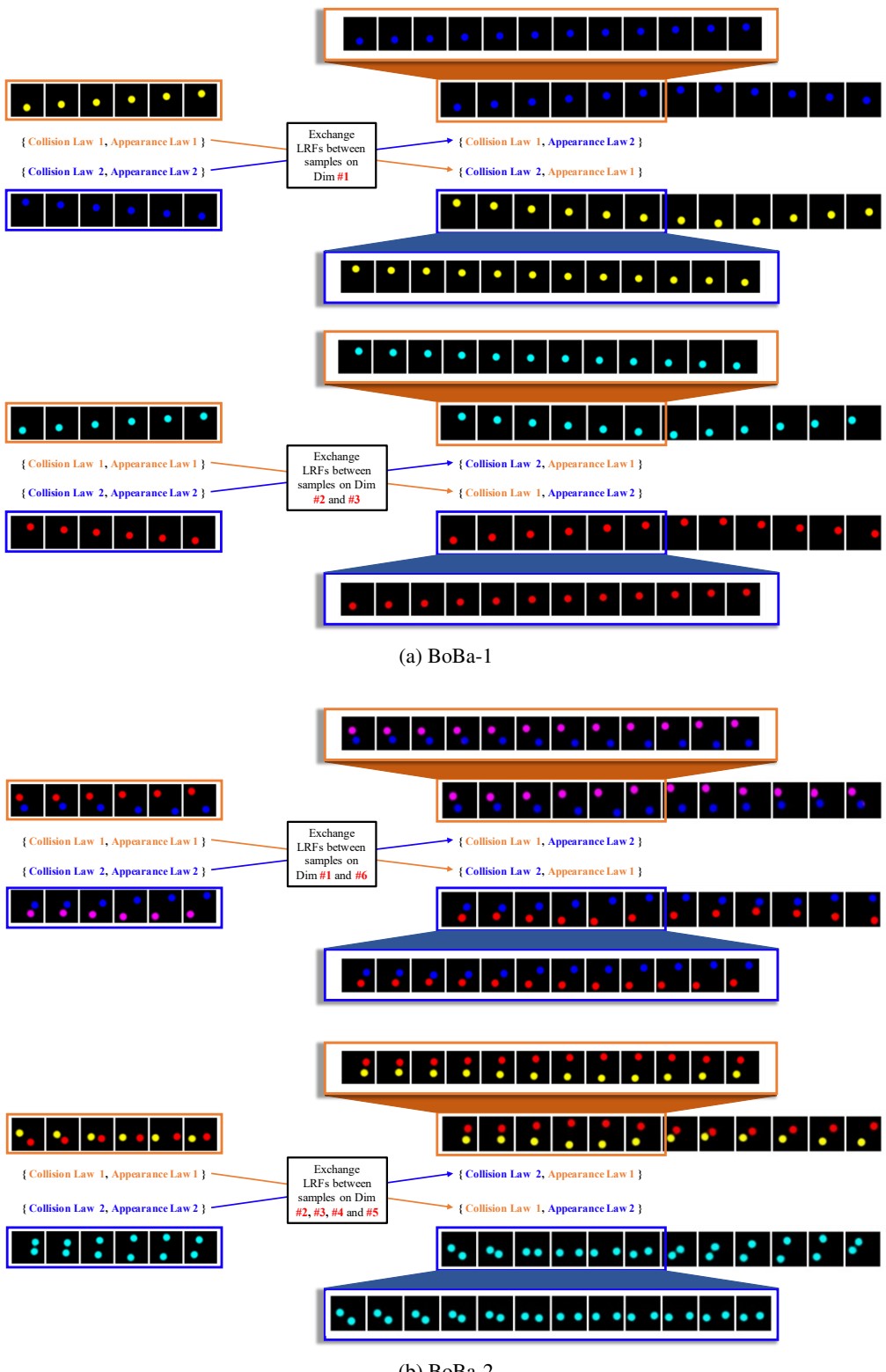

Figure 17: Law Exchange on BoBa.

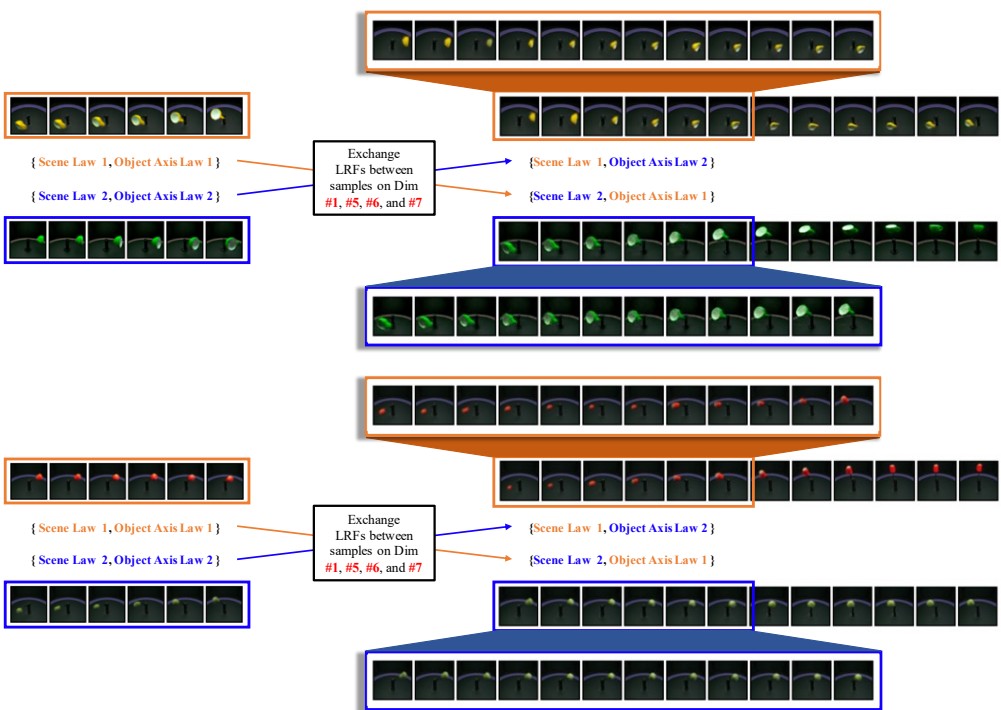

Figure 18: Law Exchange on MPI3D.

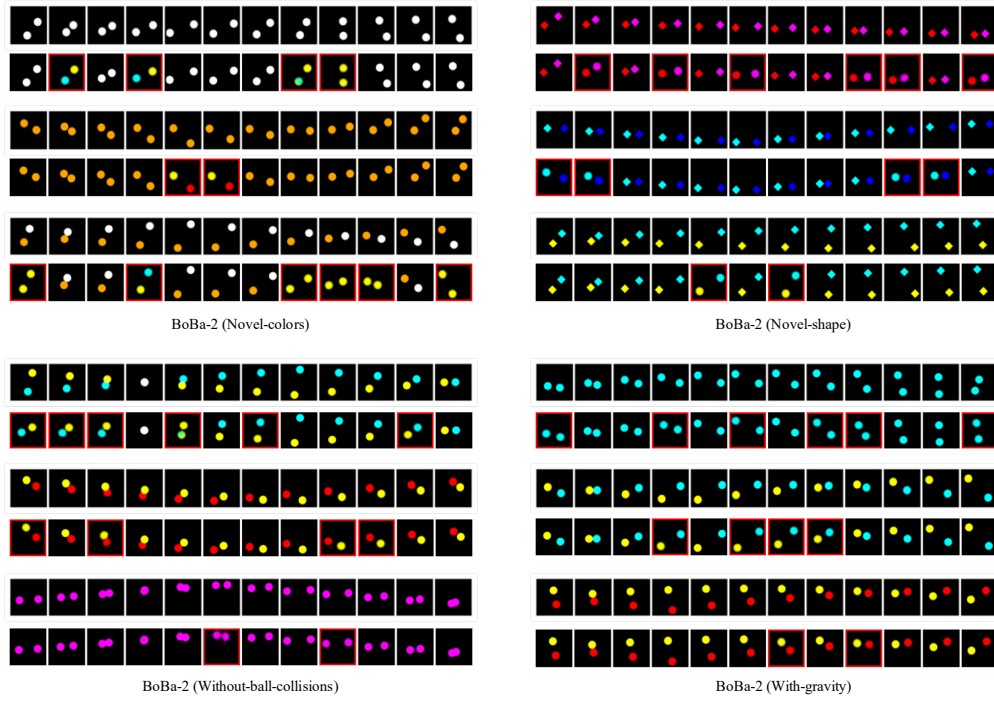

Figure 19: Evaluate CLAP-NP on BoBa-2 with novel colors, shapes, and physical laws.

Table 8: MSE and SA scores on BoBa-2 with novel concepts or laws where the models are tested with the configuration $\eta = 4/12$. SA-$k$ indicates the SA scores with $k$ candidates.

| Metric | Model | BoBa-2-NC | BoBa-2-NS | BoBa-2-WBO | BoBa-2-WG |
|---|---|---|---|---|---|
| MSE | NP | $2047.6 \pm 10.9$ | $990.2 \pm 14.1$ | $1743.1 \pm 15.7$ | $1525.5 \pm 12.9$ |
| | GP | $2671.9 \pm 75.9$ | $1907.4 \pm 74.2$ | $2068.2 \pm 124.9$ | $2019.9 \pm 85.0$ |
| | GQN | $2350.8 \pm 55.4$ | $1459.7 \pm 24.6$ | $2365.8 \pm 38.3$ | $2092.9 \pm 23.5$ |
| | CLAP-NP | $\mathbf{1611.2 \pm 71.1}$ | $\mathbf{979.0 \pm 72.5}$ | $\mathbf{1420.3 \pm 60.4}$ | $\mathbf{1322.4 \pm 58.9}$ |
| SA-2 | NP | $0.984 \pm 0.004$ | $0.989 \pm 0.003$ | $0.925 \pm 0.010$ | $0.956 \pm 0.008$ |
| | GP | $0.991 \pm 0.005$ | $0.986 \pm 0.009$ | $0.974 \pm 0.011$ | $0.985 \pm 0.012$ |
| | GQN | $0.952 \pm 0.009$ | $0.955 \pm 0.010$ | $0.878 \pm 0.015$ | $0.891 \pm 0.012$ |
| | CLAP-NP | $\mathbf{0.995 \pm 0.004}$ | $\mathbf{0.999 \pm 0.001}$ | $\mathbf{0.997 \pm 0.002}$ | $\mathbf{0.998 \pm 0.001}$ |
| SA-3 | NP | $0.972 \pm 0.005$ | $0.981 \pm 0.004$ | $0.876 \pm 0.018$ | $0.924 \pm 0.005$ |
| | GP | $0.986 \pm 0.013$ | $0.970 \pm 0.021$ | $0.940 \pm 0.035$ | $0.972 \pm 0.019$ |
| | GQN | $0.920 \pm 0.007$ | $0.926 \pm 0.012$ | $0.821 \pm 0.009$ | $0.820 \pm 0.015$ |
| | CLAP-NP | $\mathbf{0.990 \pm 0.006}$ | $\mathbf{0.998 \pm 0.002}$ | $\mathbf{0.995 \pm 0.005}$ | $\mathbf{0.998 \pm 0.002}$ |
| SA-5 | NP | $0.949 \pm 0.008$ | $0.969 \pm 0.005$ | $0.796 \pm 0.012$ | $0.877 \pm 0.010$ |
| | GP | $\mathbf{0.982 \pm 0.012}$ | $0.962 \pm 0.021$ | $0.907 \pm 0.049$ | $0.968 \pm 0.018$ |
| | GQN | $0.884 \pm 0.013$ | $0.887 \pm 0.010$ | $0.734 \pm 0.028$ | $0.741 \pm 0.013$ |
| | CLAP-NP | $0.978 \pm 0.016$ | $\mathbf{0.997 \pm 0.003}$ | $\mathbf{0.991 \pm 0.006}$ | $\mathbf{0.995 \pm 0.004}$ |
| SA-9 | NP | $0.918 \pm 0.017$ | $0.945 \pm 0.009$ | $0.705 \pm 0.021$ | $0.823 \pm 0.020$ |
| | GP | $0.954 \pm 0.031$ | $0.948 \pm 0.021$ | $0.879 \pm 0.038$ | $0.946 \pm 0.028$ |
| | GQN | $0.828 \pm 0.018$ | $0.842 \pm 0.015$ | $0.627 \pm 0.026$ | $0.643 \pm 0.028$ |
| | CLAP-NP | $\mathbf{0.975 \pm 0.010}$ | $\mathbf{0.992 \pm 0.005}$ | $\mathbf{0.982 \pm 0.008}$ | $\mathbf{0.991 \pm 0.004}$ |
| SA-17 | NP | $0.884 \pm 0.015$ | $0.917 \pm 0.011$ | $0.584 \pm 0.026$ | $0.752 \pm 0.019$ |
| | GP | $0.938 \pm 0.044$ | $0.916 \pm 0.045$ | $0.826 \pm 0.030$ | $0.926 \pm 0.034$ |
| | GQN | $0.781 \pm 0.023$ | $0.782 \pm 0.026$ | $0.549 \pm 0.014$ | $0.558 \pm 0.016$ |
| | CLAP-NP | $\mathbf{0.965 \pm 0.014}$ | $\mathbf{0.991 \pm 0.006}$ | $\mathbf{0.972 \pm 0.007}$ | $\mathbf{0.982 \pm 0.013}$ |

serve collisions between balls and borders; for BoBa-2-WG, we add vertical gravity to scenes. After training on BoBa-2, CLAP-NP is tested on four datasets without retraining to evaluate whether the compositionality of laws improves the model's generalization ability in scenes with unseen concepts or laws. Figure 19 shows that CLAP-NP can predict the correct object positions on BoBa-2-NC and BoBa-2-NS. CLAP-NP predicts inaccurate color and shape of objects because the encoder and decoder are trained on BoBa-2, and the latent space does not encode the unseen colors or shapes. We observe similar results on BoBa-2-WBO and BoBa-2-WG that CLAP-NP learns the correct law of appearance but predicts incorrect positions of balls when there are unseen physical laws in scenes. To better evaluate the generalization ability in scenes with novel laws, we quantitatively evaluate CLAP-NP and baseline models with MSE ad SA scores on the four datasets. The results in Table 8 indicate that CLAP-NP achieves the best MSE and SA scores in most situations, which illustrates CLAP-NP's generalization ability in scenes with novel concepts or laws.

### E.8 NUMBER OF LATENT RANDOM FUNCTIONS

This experiment explores the influence of setting too many or too few LRFs in CLAP-NP. Figure 20 shows the latent traversal results on MPI3D, BoBa-2, and CRPM-DT. If we set too few LRFs, CLAP-NP encodes different laws in one LRF instead of learning compositional laws, which will influence CLAP-NP's generation ability (e.g., set only two LRFs for BoBa-2). Setting too many LRFs has no significant influence on CLAP-NP's performance because there will be redundant dimensions in CLAP-NP that do not encode information (e.g., dimensions 1, 3, and 6 on CRPM-DT). However, due to the independent modeling of concept-specific LRFs, setting a large number of LRFs will reduce the computational speed of CLAP-NP and increase the number of model parameters.

### E.9 PREDICTION STRATEGY

CLAP uses the one-shot strategy that predicts all target images at one time. However, the rollout strategy can be another choice to predict the following target images through the few context images

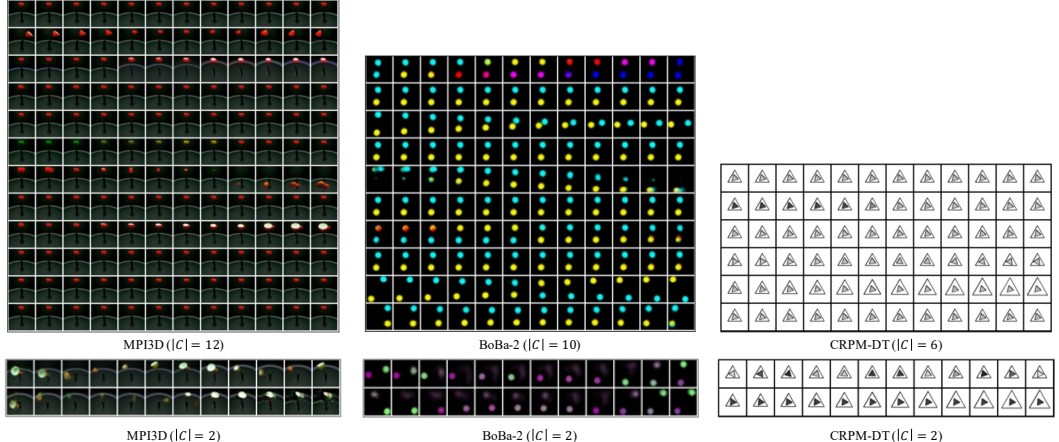

Figure 20: Latent traversal results with too many (top) and too few (bottom) LRFs.

Table 9: MSE scores of CLAP-NP with the rollout and one-shot strategy.

| Dataset | CRPM-DT ($\eta = 6/9$) | BoBa-2 ($\eta = 6/12$) | MPI3D ($\eta = 20/40$) |
|---------|------------------------|------------------------|------------------------|
| Rollout | $362.5 \pm 3.5$ | $\mathbf{4651.8 \pm 15.7}$ | $\mathbf{228.2 \pm 0.6}$ |
| One-shot | $\mathbf{361.9 \pm 2.3}$ | $4778.5 \pm 15.8$ | $237.3 \pm 1.2$ |

at the beginning. For example, in the test configuration $\eta = 20/25$, we first predict the 6th to 10th images from the first five context images, then combine them (1st to 10th images) to predict the following five images, and repeat this process until all target images are predicted. Table 9 shows the MSE scores on CRPM-DT, BoBa-2, and MPI3D where CLAP-NP predicts targets with the rollout and one-shot strategy, respectively. On BoBa-2 and MPI3D, the rollout strategy slightly improves the prediction accuracy. Generally, if a target image is far away from all the context images, the prediction may have high uncertainty. The model only predicts a few target images close to the context each time with the rollout strategy, while the one-shot strategy requires the model to predict all the target images at one time. Therefore, the rollout strategy is more likely to have lower computational uncertainty and higher prediction accuracy.

### E.10 Failure Cases

In this experiment, we display some failure cases. For BoBa, most failure cases occur when there are continuous target images (1st sample of BoBa in Figure 21) or too many target images (2nd sample of BoBa in Figure 21). For CRPM, CLAP can predict target images with diversity when we remove a row from a matrix, but it sometimes generates target images that break the rules. For example, the color of images keeps invariant in the first two rows, but CLAP generates images in the third row with changing grayscales (samples of CRPM in Figure 21). For MPI3D, when the object size has an obvious change (1st sample of MPI3D in Figure 21) or the centric object is tiny (2nd sample of MPI3D in Figure 21), the predictions can be incorrect or unclear.

### F Limitations

We conclude our limitations in two aspects. (1) **Complexity of datasets.** Because compositional law parsing is an underexplored task, we first validate the effectiveness of CLAP on datasets with relatively clear and simple rules to avoid the influence of unknown confounders in complicated datasets. We believe that the compositionality of laws also exists in more complex scenarios (e.g., learning physical laws in realistic scenes) and some vision tasks may benefit from compositional law parsing. For example, we can perform a controllable video generation process based on law modification or make more interpretable predictions by analyzing dominant laws in videos. Discovering such compositional law parsing ability in more complex situations can be a valuable topic in future works.

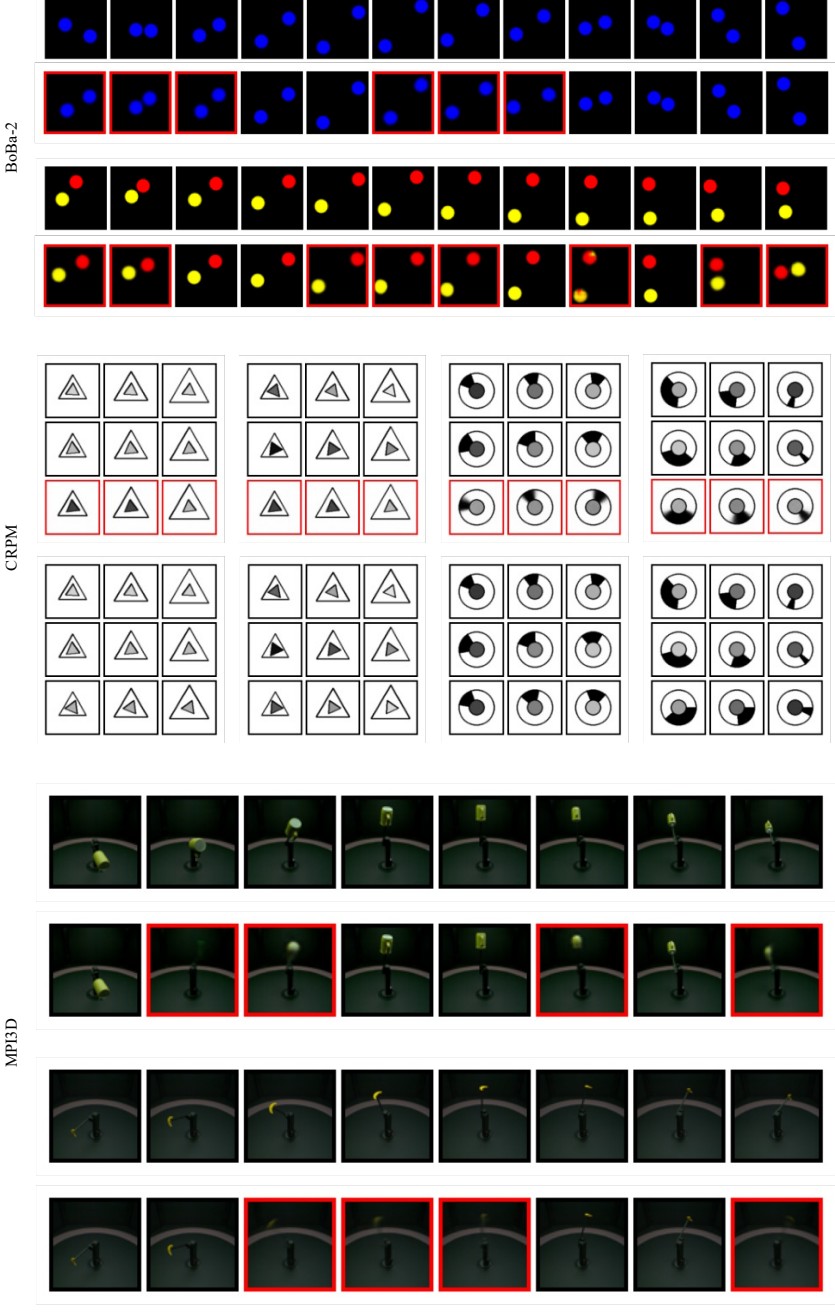

Figure 21: Failure cases of CLAP-NP.

(2) **Setting the number of LRFs.** CLAP places an upper bound on the number of LRFs. For scenes with multiple complex rules, we can empirically set an appropriate upper bound or directly put a large upper bound for the number of LRFs. However, a large bound will linearly increase the number of model parameters, and the redundant concepts will waste computing resources, decreasing computational efficiency. Therefore, exploring a mechanism in the future to dynamically adjust the number of functions will be meaningful. When applying CLAP to more complex scenes, we may introduce more inductive biases for better concept decomposition (e.g., use a more task-specific encoder or decoder). We design CLAP as an unsupervised model, which means that it can be extended to take advantage of task-specific annotations. Thus we can find ways to integrate CLAP with additional supervision information (e.g., supervise CLAP with the changing factors of scenes) to help concept learning in a specific task.

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
