# OpenReview forum: "Compositional Law Parsing with Latent Random Functions"
_ICLR.cc/2023/Conference — ICLR 2023 poster_

### Official Review · Reviewer_4jVq · 2022-10-24

**Confidence:** 2
**Clarity, Quality, Novelty And Reproducibility:** 1. Paper is clear and concise.
2. Hig…
**Correctness:** 4
**Technical Novelty And Significance:** 4
**Empirical Novelty And Significance:** 3
**Recommendation:** 8

**Strength And Weaknesses:**

Strengths
1. Paper is well-written and well-motivated.
2. The concepts are clearly explained with formulas and diagrams
3. Claims are sufficiently supported by experiments.

Weaknesses
1. Paper introduces several concepts which are not mainstream. However, this is rectified by providing the reader with references and extensive Appendix material.

**Summary Of The Paper:**

Paper proposes a novel method to decompose a scene into representation of concepts such as object appearance, background, and angle of object rotation. This is done through an encoding-decoding architecture which represents the concepts as latent variables. These concepts are then recomposite via a decoder network to generate the target images.

Paper further proposes to use Neural Processes (CLAP-NP) to instantiated each learnt concepts for inference purposes. Different experiments were ran to validate the model performance against baseline models for the intuitive physics, abstract visual reasoning and scene representation tasks. The proposed method significantly overperforms all baseline methods with the MSE and new derived metric, Selection Accuracy (SA).

Finally, paper also shows that the method can be used to edit the individual concepts to generate new samples.

**Summary Of The Review:**

Overall, this work seems excellent and of high impact. However, due to my unfamiliar with the technique described, it is difficult for me to review this work properly.

The experimental results are convincing. But I note that there is no comparison with any SOTA methods for tasks.

---

> ### Author Response · Authors · 2022-11-18
> **Response to Reviewer 4jVq**
>
> Thanks for the constructive suggestions. The revised version of our submission has been uploaded according to the suggestions. The detailed responses to the reviewer's comments are as follows
>
>
>
> **Q1. Lack SOTA models**
>
> > ... But I note that there is no comparison with any SOTA methods for tasks.
>
> Realizing compositional law parsing through random functions is an underexplored problem. As far as we know, no existing compositional law parsers can be used as SOTA models for comparison. Thus we only chose non-compositional random functions (e.g., NP and GQN) as baseline models. By comparing with the baselines, our experimental results illustrate the motivation of CLAP to implement compositional law parsing, as well as law manipulation and better generalization ability in scenes with novel laws.

---

### Official Review · Reviewer_rEh1 · 2022-10-25

**Confidence:** 3
**Correctness:** 4
**Technical Novelty And Significance:** 3
**Empirical Novelty And Significance:** 3
**Recommendation:** 6

**Clarity, Quality, Novelty And Reproducibility:**

Overall the writing is easy to follow and the figures are carefully drawn. No specific concerns, but here are a couple of suggestions:
- __On choice of notation__: I found it confusing to think about concepts A and context C. It would be easier if concepts were denoted C, and if you used S for the source/context frames, used to predict targets T.
- It would be useful to have a __summary table__ describing exactly what $x_t$ is for each dataset (e.g. timestep for BoBA, azimuth/altitute for MPI3D).


**Strength And Weaknesses:**

Strengths:
- Multiple datasets with different focus
- Illustration of compositionalilty
- Demonstration of failure cases

Cons:
- Choice of baseline: neither NP nor GQN seem like strong, obvious baselines. Wouldn’t a system like [Aloe (Ding et al. 2020)](https://arxiv.org/abs/2012.08508) offer a better comparison?
- Based on Tables 3 and 4, I worry that the model needs careful tuning to work or several runs for a winner to emerge. The best hyperparameters vary significantly across datasets.

**Summary Of The Paper:**

The paper proposes a latent variable model to capture concept-specific laws from images/videos. The laws are encoded as Latent Random Functions (specifically, Neural Processes), and can be composed with one another.

**Summary Of The Review:**

The following questions will help me assess the work better:
- Is anything preventing you from __assessing__ the __disentanglement__ of the LRFs? It would be useful to have those metrics (e.g. see https://github.com/google-research/disentanglement_lib).
- What would be __other potential realizations__ of CLAP which don’t use Neural Processes?
- __Future work__: how do you expect LRFs would perform on the [Abstraction and Reasoning Challenge](https://www.kaggle.com/c/abstraction-and-reasoning-challenge)? Would that be an ideal target domain?
- __How to generate future predictions__: You could generate an extended rollout (e.g. on BoBa) by applying the model autoregressively on the most recent generated frames. For example: to generate 20 target frames after 5 context frames, you could first generate frames 6-10, then use those to generate 11-15, etc. Alternatively, you could evaluate one-shot using a \eta=20/25. Which one do you expect would work better? The former strategy would allow you to re-infer the concepts and global latent at each application.
- On specific figures:
  - Figure 7: The newly generated example on CRPM-T doesn’t have the same color law as matrix 3 right? I expected the triangles in the top row to be dark and identically colored. Rather, the triangles in the top row seem to be following a pattern that is different from all other top rows (from the context image). How is that possible?
  - Figures 4 and 5: could you confirm you’re using a single model on each dataset (but multiple test configurations)?
  - Figure 14: do you learn any latents on CRPM-DT which change the size/rotation of the outer triangle?

---

> ### Author Response · Authors · 2022-11-18
> **Response to Reviewer rEh1 (Part II)**
>
> **Q7. Future work on ARC**
>
> > how do you expect LRFs would perform on the [Abstraction and Reasoning Challenge](https://www.kaggle.com/c/abstraction-and-reasoning-challenge)? Would that be an ideal target domain?
>
> We think that CLAP can be potentially applied to ARC. We can regard a task on ARC as a function mapping the input images to output images. In this case, the inputs and outputs will satisfy a task-specific changing pattern (e.g., fill the color in the specific region of input images), which can be represented as a law in CLAP. We can use CLAP to decompose the law into sub-laws represented by latent random functions (e.g., fill the region with yellow and keep the border color unchanged, respectively). CLAP can handle novel tasks through the combination of the learned sub-laws.
>
>
>
> **Q8. Comparison of the rollout and one-shot prediction strategy**
>
> > You could generate an extended rollout (e.g. on BoBa) by applying the model autoregressively on the most recent generated frames. For example: to generate 20 target frames after 5 context frames, you could first generate frames 6-10, then use those to generate 11-15, etc. Alternatively, you could evaluate one-shot using a \eta=20/25. Which one do you expect would work better? The former strategy would allow you to re-infer the concepts and global latent at each application.
>
> Thanks for the suggestion. We have conducted the experiment of the suggested rollout and our one-shot prediction strategies on the CRPM-DT, BoBa-2, and MPI3D datasets. The MSE scores of two prediction strategies are
>
> | Dataset  | CRPM-DT ($\eta=6/9$) | BoBa-2 ($\eta=6/12$)  | MPI3D ($\eta=20/40$) |
> | :------: | :------------------: | :-------------------: | :------------------: |
> | Rollout  |   362.5 $\pm$ 3.5    | **4651.8 $\pm$ 15.7** | **228.2 $\pm$ 0.6**  |
> | One-shot | **361.9 $\pm$ 2.3**  |   4778.5 $\pm$ 15.8   |   237.3 $\pm$ 1.2    |
>
> We observe that the rollout strategy slightly improves the MSE scores on BoBa-2 and MPI3D. In the test configuration of this experiment, we only give the first few frames of the sample and let CLAP predict the following target images. The rollout strategy predicts only a few target images close to the context images each time, while the one-shot strategy requires the model to predict target images far away from the given context images. Therefore, the rollout strategy is more likely to reduce the uncertainty in prediction and increase prediction accuracy. We have **added this experiment** to Appendix D.9 in the revised version.
>
>
>
> **Q9. Color deviation in the generated CRPM-T sample of Figure 7**
>
> > Figure 7: The newly generated example on CRPM-T doesn’t have the same color law as matrix 3 right? I expected the triangles in the top row to be dark and identically colored. Rather, the triangles in the top row seem to be following a pattern that is different from all other top rows (from the context image). How is that possible?
>
> We think that the slight color deviation of the generated CRPM-T sample in Figure 7 is mainly caused by the sampling of concepts. Assuming that the latent random function of color we used to generate the new sample is $f^{clr}$, we will sample $\boldsymbol{z}^{clr}\_{n} \sim p(\boldsymbol{z}^{clr}\_{n}|f^{clr},\boldsymbol{x}^{clr}\_{n})$ to generate the concept of color for the $n$th image. The randomness of the sampling process may cause the color concept of the new sample to deviate from that of the original sample, which finally leads to the deviation in pixels.
>
>
>
> **Q10. Details about the model and experiments**
>
> > Figures 4 and 5: could you confirm you’re using a single model on each dataset (but multiple test configurations)?
>
> Yes. Both Figure 4 and Figure 5 display the prediction results of a single model under different test configurations.
>
> > Figure 14: do you learn any latents on CRPM-DT which change the size/rotation of the outer triangle?
>
> CLAP-NP learns the latent random function on the size of the outer triangle but does not learn that on the rotation of the outer triangle. In CRPM-DT, the rotation of the outer triangle does not change; the law of rotation is displayed in the inner triangle.

---

> ### Author Response · Authors · 2022-11-18
> **Response to Reviewer rEh1 (Part I)**
>
> Thanks for the constructive suggestions. We have conducted the experiment and discussion on different prediction strategies, made modifications to the notations, and added a summary table to describe $\boldsymbol{x}_{t}$ according to the comments. The revised version of our submission has been uploaded. The detailed responses to the reviewer's comments are as follows
>
>
>
> **Q1. Concerns on baselines**
>
> > Choice of baseline: neither NP nor GQN seem like strong, obvious baselines. Wouldn’t a system like [Aloe (Ding et al. 2020)](https://arxiv.org/abs/2012.08508) offer a better comparison?
>
> Thanks for the suggestion. We chose the non-compositional random functions NP and GQN as baselines to better illustrate our motivation to introduce compositionality into law parsing. Aloe (Ding et al. 2020) does not **explicitly** model the changing laws among frames (e.g., NP and GQN explicitly represent the changing laws by random functions). Instead, the changing patterns are implicit in the parameters of black-box neural networks. For example, when learning the laws in the samples from CRPM-DT (Figure 5), we can hardly find the explicit representation of the color-changing pattern from Aloe, making it difficult for Aloe to understand how the underlying laws constitute the entire changing pattern of a scene.
>
>
>
> **Q2. Concerns on hyperparameter tuning**
>
> > Based on Tables 3 and 4, I worry that the model needs careful tuning to work or several runs for a winner to emerge. The best hyperparameters vary significantly across datasets.
>
> First, we would like to clarify that Table 4 (now Table 5 in the revised version) indeed shows the choice of hyperparameters in the baseline NP. Table 3 (now Table 4 in the revised version) shows the choice of hyperparameters for CLAP-NP. Since CLAP-NP is trained in an unsupervised manner, explicit inductive biases in the model and data are necessary for concept learning [1]. The architecture and hyperparameters of the model can be regarded as inductive biases. CLAP-NP uses NP as latent random functions on all three datasets without task-specific networks and does not takes advantage of additional supervision. Therefore, it is necessary for CLAP-NP to set different hyperparameters as inductive biases to handle datasets with obvious differences.
>
> [1] Locatello, Francesco, et al. "Challenging common assumptions in the unsupervised learning of disentangled representations." *international conference on machine learning*. PMLR, 2019.
>
>
>
> **Q3. Choice of notation**
>
> > I found it confusing to think about concepts A and context C. It would be easier if concepts were denoted C, and if you used S for the source/context frames, used to predict targets T.
>
> We have changed these notations in the revised version. Thanks for the suggestion.
>
>
>
> **Q4. Summary table**
>
> > It would be useful to have a summary table describing exactly what xt is for each dataset (e.g. timestep for BoBA, azimuth/altitute for MPI3D).
>
> We have added Table 3 to describe $\boldsymbol{x}_{t}$ for each dataset in the revised version. Thanks for the suggestion.
>
>
>
> **Q5. Disentanglement metric**
>
> > Is anything preventing you from **assessing** the **disentanglement** of the LRFs? It would be useful to have those metrics (e.g. see https://github.com/google-research/disentanglement_lib).
>
> As shown in Figure 6, the physical laws and the appearance-invariant laws are obtained by combining latent random functions (LRFs). Each LRF represents a sub-law, e.g., the third dimension encodes the position change on the x-axis, and the fifth dimension encodes the position change on the y-axis. These sub-laws constitute the entire physical law on the position of balls. However, these sub-laws are automatically learned from CLAP-NP, and there are **no** ground truth labels of these sub-laws in samples. Moreover, the way to decompose some laws into LRFs is not unique (e.g., one can decompose the motion of balls in different ways), so there are **no** ground truths for LRFs in this case. The computation of the disentanglement metric requires us to know the changing factors of each sample, so it is difficult to compute the disentanglement metric on LRFs.
>
>
>
> **Q6. Other realizations of CLAP**
>
> > What would be other potential realizations of CLAP which don’t use Neural Processes?
>
> Other random functions from the NP family can also instantiate CLAP (e.g., Attentive NP [1] and Convolutional NP [2]). The common point of these methods is to describe the function space by explicitly defined probability distributions (such as NP's Gaussian distributed global latent variables), which can help us find potential candidates for latent random functions.
>
> [1] Kim, Hyunjik, et al. "Attentive Neural Processes." *International Conference on Learning Representations*. 2019.
>
> [2] Foong, Andrew, et al. "Meta-learning stationary stochastic process prediction with convolutional neural processes." *Advances in Neural Information Processing Systems* 33 (2020): 8284-8295.

---

### Official Review · Reviewer_Fbxt · 2022-10-26

**Confidence:** 4
**Correctness:** 4
**Technical Novelty And Significance:** 3
**Empirical Novelty And Significance:** 3
**Recommendation:** 6

**Clarity, Quality, Novelty And Reproducibility:**

- Clarity
    - The paper is generally well written and easy to follow. The figures are of high quality.
- Quality
    - The evaluation makes sense, and there are some interesting results.
- Novelty
    - The novelty compared to ROOTS is not sufficiently discussed.
- Reproducibility
    - Dataset, architecture details and hyperparameters are provided in Appendix.

**Strength And Weaknesses:**

- Strengths
    - The experiment results are quite extensive. The failure of NP and GQN is clearly shown. The law manipulation result (Figure 7) is very interesting and shows that the learned neural processes can indeed be composed.
    - The paper is well organized, and introduces sufficient background knowledge for readers to follow.
- Weaknesses
    - At a high level, the proposed method is similar to [ROOTS](https://www.jmlr.org/papers/v22/20-1176.html), which brings compositionality into GQN and focuses on compositional understanding of 3D scenes. Both of them can be thought of as a compositional NP framework. ROOTS has similarly shown better generation quality than GQN, and can compose novel scenes by combining the object-level GQNs. However, I think there is still merit in this paper, as it seems more general and not limited to 3D scenes. I encourage the authors to highlight their novelty compared to ROOTS.
    - The datasets look visually simple. Some compositional video scene understanding methods (e.g., Kosiorek et al., 2018; Jiang et al., 2019; Lin et al., 2020a as listed in related work) can deal with more complex scenes. Can you clarify why they are not considered baselines?
- Questions and Minor Issues
    - How many concepts did you use and what is the dimension of the latent vector for each concept?
    - The bouncing ball dataset is sometimes referred to as BoBa and sometimes BaBo.

**Summary Of The Paper:**

The paper brings compositionality into the Neural Process framework. It decomposes the input into multiple concepts, each represented by a latent vector. A Neural Process is learned for each concept to capture its law, so that the concept can be generated at query points given a few context points. The full pipeline is trained end-to-end via variational inference without supervision. Experiments on three diverse datasets show that the proposed model can produce more accurate pixel predictions and more consistent latent space than NP and GQN. It is also qualitatively shown that one can compose different laws by combining the corresponding neural processes.

**Summary Of The Review:**

I am leaning toward reject for the paper in its current form, mainly because the novelty needs more clarification.

---

> ### Author Response · Authors · 2022-11-18
> **Response to Reviewer Fbxt**
>
> Thanks for the constructive suggestions. The revised version of our submission has been uploaded. The detailed responses to the reviewer's comments are as follows
>
>
>
> **Q1. Highlight the novelty compared to ROOTS**
>
> > At a high level, the proposed method is similar to ROOTS, which brings compositionality into GQN and focuses on compositional understanding of 3D scenes. Both of them can be thought of as a compositional NP framework. ROOTS has similarly shown better generation quality than GQN, and can compose novel scenes by combining the object-level GQNs. However, I think there is still merit in this paper, as it seems more general and not limited to 3D scenes. I encourage the authors to highlight their novelty compared to ROOTS.
>
> Actually, CLAP and ROOTS try to solve different problems. ROOTS captures the **compositionality of objects** in 3D scenes (i.e., learning object-centric representations), while CLAP learns the **compositionality of laws** (e.g., in Figure 4, CLAP-NP can parse the collisions of objects in a scene). In ROOTS, one can edit object-centric representations to obtain 3D scenes with different objects; in CLAP, we modify latent random functions to acquire samples satisfying novel laws.
>
> Taking the CRPM-DT dataset (Figure 5) as an example, ROOTS can obtain the representation of the inner and outer triangles for each panel and the correspondence between triangles in the panels of an RPM matrix. However, ROOTS cannot learn changing rules on attributes (such as rotation and color of the triangle) because it models how objects in the scene are composed but cannot understand the concept-changing rules and how the rules constitute the general law of scenes.
>
>
>
> **Q2. Why not use compositional video scene understanding methods as baselines**
>
> > The datasets look visually simple. Some compositional video scene understanding methods (e.g., Kosiorek et al., 2018; Jiang et al., 2019; Lin et al., 2020a as listed in related work) can deal with more complex scenes. Can you clarify why they are not considered baselines?
>
> Thanks for the suggestion. In our opinion, compositional video scene understanding methods differ from CLAP in the following aspects:
>
> **Different Motivations.** Compositional video scene understanding methods are based on the compositionality of objects. They aim to parse individual objects in the video of scenes, learn consistent object-centric representations among frames, and predict the position of objects in future frames. CLAP learns compositional laws, decompose the changing laws in the scene into different sub-laws, and enables us to manipulate specific laws on samples. And CLAP can also be used to model laws on other types of data, such as RPMs (Figure 5).
>
> **Different Problem Settings.**  The compositional video scene understanding methods predict the following frames discretely and frame-by-frame (i.e., given frames at $t=1$ and $t=2$ to predict the frame at $t=3$). CLAP uses random functions to **explicitly model the changing laws** in the entire video. Thus we can predict the images in continuous intervals through the given images at any time (i.e., given frames at $t=1.5$ and $t=3.4$ to predict frames at $t=2.1$ and $t=8.3$).
>
> Compositional video scene understanding methods do not explicitly model changing laws (e.g., describe laws as random functions), so they cannot explicitly represent and manipulate the laws in the video of scenes. Therefore, we had not considered compositional video scene understanding methods as baselines in experiments.
>
>
>
> **Q3. Details about the concepts**
>
> > How many concepts did you use and what is the dimension of the latent vector for each concept?
>
> The hyperparameter $|C|$ in Table 4 indicates the maximum number of concepts we used, which is
>
> | BoBa-1 | BoBa-2 | CRPM-T | CRPM-DT | CRPM-C | CRPM-DC | MPI3D |
> | :----: | :----: | :----: | :-----: | :----: | :-----: | :---: |
> |   3    |   6    |   3    |    3    |   2    |    3    |   7   |
>
> where the dimension of the latent vector for each concept is 1.
>
>
>
> **Q4. Typos**
>
> > The bouncing ball dataset is sometimes referred to as BoBa and sometimes BaBo.
>
> Thanks for pointing out the typos. We have fixed them in the revised version.

---

> > ### Comment · Reviewer_Fbxt · 2022-12-09
> > **Thanks for your detailed response**
> >
> > My concerns have been adequately addressed. I am happy to recommend acceptance.

---

### Official Review · Reviewer_HFjB · 2022-10-26

**Confidence:** 4
**Correctness:** 3
**Technical Novelty And Significance:** 3
**Empirical Novelty And Significance:** 3
**Recommendation:** 6

**Clarity, Quality, Novelty And Reproducibility:**

The paper is well presented and the appendix includes details of datasets, hyperparameters, and network architecture. The author also submitted source code as supplemental material for reproducibility.

The empirical evaluation is based on simple datasets. It will be beneficial to the community if the authors can discuss the limitation of the proposed model, e.g. what can be represented as compositional laws and what is needed if we apply this approach to more complex scenes.


**Strength And Weaknesses:**

Strength
- This is an interesting paper that focuses on learning disentangled laws to interpret dynamic scenes.
- This paper also proposes a novel approach to learning concept-specific laws with Neural Proces.
- The paper shows several qualitative examples to show how the results are interpretable and the composition of learned latent functions is meaningful.

Weaknesses
- It is unclear how the authors determined the number of laws for learning. What if we choose too many/little laws? Or how many laws we may need to interpret a realistic video?
- The current quantitative evaluation focuses on MLE and selection accuracy to show the improvement of semantic error but these two don’t really represent the model learns the right laws. The MLE can be a result of better visual encoding, the selection accuracy shows better representation. If the model learns compositional laws, it can generalize better to novel combinations of concepts that are not in training, e.g. new colors. The paper currently only shows one generalization example in the appendix. The evaluation will be much stronger to show a quantitative evaluation of generalization capability.


**Summary Of The Paper:**

This paper proposes a latent variable model that parses dynamic scenes into independent laws represented as neural random functions. The model is trained to reconstruct the image sequence while keeping the distribution of concepts and random functions not changing over time and concepts. The evaluation on various datasets shows that it can predict the target images well. The qualitative examples show that the decomposed laws correlate with the changing visual concepts and can be swapped to generate novel samples.

**Summary Of The Review:**

The paper proposes a novel approach to interpret dynamic scenes. It is well presented and easy to follow. It provides several examples of learned compositional laws. It will be much convincing if the authors can provide quantitative evaluation of compositional generalization.

---

> ### Author Response · Authors · 2022-11-18
> **Response to Reviewer HFjB**
>
> Thanks for the constructive suggestions. We have added experiments to illustrate the effect of the number of laws and evaluate the model's generalization ability. The revised version of our submission has been uploaded. The detailed response to the reviewer's comments are as follows
>
>
>
> **Q1. How to determine the number of laws**
>
> > It is unclear how the authors determined the number of laws for learning. What if we choose too many/little laws? Or how many laws we may need to interpret a realistic video?
>
> The hyperparameter |C| in Table 4 sets the maximum number of latent random functions, thereby controlling the number of laws learned by CLAP-NP. We have **added additional experiments** (see Appendix D.8) to show the effect of too large or too small |C| on CLAP-NP. If |C| is too small, CLAP-NP confounds different laws in the same dimension. In this case, it is difficult for CLAP-NP to learn compositional laws, and the performance of reconstruction and prediction may decline. If |C| is too large, CLAP will generate redundant dimensions that do not influence the performance. For a realistic video, we can set a large |C| and gradually reduce |C| according to the number of redundant dimensions. In our future work, we can also introduce Bayesian nonparametric methods to dynamically select the number of latent random functions to reduce the impact of redundant dimensions on computational speed.
>
>
>
> **Q2. Quantitative Evaluation of Generalization Capability**
>
> > The current quantitative evaluation focuses on MLE and selection accuracy to show the improvement of semantic error but these two don’t really represent the model learns the right laws. The evaluation will be much stronger to show a quantitative evaluation of generalization capability.
>
> The MSE and SA scores in Table 1 and Figure 3 have demonstrated the generalization ability of CLAP-NP in samples containing more target images (e.g., on BoBa-2, we train models with $\eta=1 \sim 4/12$ and test them with $\eta=6/12$). To better illustrate the generalization ability of models in scenes with unseen concepts or laws, we have added **an additional experiment** in Appendix D.7. Based on BoBa-2, we generate four datasets with unseen object colors, with unseen object shapes, without collisions between objects, and with vertical gravity to test the model's generalization ability. We quantitatively measure the generalization ability in MSE and SA scores on the four datasets where CLAP-NP achieves the best MSE and SA scores in most situations. The detailed experimental results are shown in Table 7.
>
>
>
> **Q3. Add Limitations**
>
> > The empirical evaluation is based on simple datasets. It will be beneficial to the community if the authors can discuss the limitation of the proposed model, e.g. what can be represented as compositional laws and what is needed if we apply this approach to more complex scenes.
>
> Thanks for the suggestion. Introducing compositionality into random functions is an underexplored task. We consider datasets with relatively clear and simple rules here because we want first to validate the effectiveness of SVCL and prevent the influence of unknown confounders in complex scenes. We have added a brief discussion about the limitations in the final section of the main text and a more detailed discussion of the limitations and future works in Appendix E.

---

### Official Review · Reviewer_i5ZN · 2022-10-31

**Confidence:** 2
**Correctness:** 3
**Technical Novelty And Significance:** 4
**Empirical Novelty And Significance:** 4
**Recommendation:** 6

**Clarity, Quality, Novelty And Reproducibility:**

## Clarity
The figures are clear and the concept of the paper is understandable. The presentation, however, could be improved: the abstract is not explicit enough to ground the main concepts, sections 3 and 4 are detailed but lack simpler explanations to tie together the main concepts.

## Quality
The quaility of the paper is high. The figures, mathematical formulations and experiments are laid out with care. The overall detail of the paper is high and the framework proposed appears to be carefully crafted and quite powerful.

## Novelty
The ideas appear to be generally novel.

## Reproducibility
The paper appears hard to reproduce from the manuscript alone. Great care is taken to ensure that the mathematical framework is explained in detail, but specific implementation procedures and explanations related to experimental details are lacking.

**Strength And Weaknesses:**

## Strengths
- Powerful concept with several meaningful abstractions
- In-depth mathematical formulation explaining the formalities of the proposed framework
- Detailed experiments, including multiple different benchmarks with very different setups and objectives
- Very strong performance against baselines

## Weaknesses
- The abstract is not very good at exemplifying the proposed framework. More information, and possibly examples, describing what each law and concept can be would be great so that the concept becomes clear from the get-go.
- The explanation of the Neural Process in Section 3 is hard to understand. Clarifying that paragraph might be of great help to readers unfamiliar with NPs.
- General clarity. The paper is not easy to follow without substantial previous knowledge of the field, and it is hard to judge the
- Baselines used: the set of baselines used is small, and an inclusion of other works could improve the experiments section. Importantly, comparing to task-specific networks might be interesting to observe how this general framework competes.


**Summary Of The Paper:**

The paper proposes a model for compositional law parsing that is capable to accurately represent different "laws" of semantic concepts. The laws are built with random functions, and the proposed system can combine these functions to achieve flexibility and generality in different law parsing tasks.

The model uses an encoder-decoder architecture that represents concepts as latent variables, and laws are operationalized as concept-specific functions built with neural processes.

The authors claim that the proposed framework outperforms other methods in scene representation, intuitive physics and abstract visual reasoning tasks.

**Summary Of The Review:**

The paper introduces a novel and powerful concept. The experiments are thorough and the mathematical formulation appears to be thoroughly supported. Its main weakness is tied to clarity: the manuscript assumes large amount of previous knowledge related to NP and other topics, and clarity could be improved to ensure more fluid reading. In general, the paper establishes a powerful framework with strong improved performance over baselines over several benchmarks.

---

> ### Author Response · Authors · 2022-11-18
> **Response to Reviewer i5ZN**
>
> Thanks for the constructive suggestions. We have modified the abstract and Preliminaries (Section 3) to improve the paper's clarity and added GP with deep kernels as a new baseline. The revised version of our submission has been uploaded according to the suggestions. The detailed responses to the reviewer's comments are as follows
>
>
>
> **Q1. Improving the clarity of the paper**
>
> > The abstract is not very good at exemplifying the proposed framework. More information, and possibly examples, describing what each law and concept can be would be great so that the concept becomes clear from the get-go.
>
> > The explanation of the Neural Process in Section 3 is hard to understand. Clarifying that paragraph might be of great help to readers unfamiliar with NPs.
>
> Thanks for the suggestions. To improve the abstract, we added "For example, CLAP learns the laws of position-changing and appearance constancy from the moving balls in a scene, making it ..." as an example at the end of the abstract to better illustrate CLAP's ability of compositional law parsing. We have also revised the Preliminaries (Section 3) to provide a more easy-to-understand introduction to random functions and NPs.
>
>
>
> **Q2. The use of baselines**
>
> > the set of baselines used is small, and an inclusion of other works could improve the experiments section. Importantly, comparing to task-specific networks might be interesting to observe how this general framework competes.
>
> Thanks for the suggestion. Our goal is to achieve compositional law parsing through latent random functions. Therefore, using non-compositional random functions as baselines can better illustrate the motivation and capability of CLAP to introduce compositionality into law parsing. To enlarge the set of baselines, we have added **GP with deep kernels** as a baseline, which can be a representation of kernel-based random functions. **Additional experimental results** about GP are added in Table 1, Figures 3-5 of the main text, and Table 7, Figures 9-13 in the appendix. Details about GP can be found in Appendix C.2. In the updated experiments. GP performs better when we provide more context points of a scene (e.g., in MPI3D, when we give 30 context images, GP achieves better MSE scores than NP and GQN). But in general, CLAP-GP outperforms GP on all datasets.
>
>
>
> **Q3. Reproducibility of the paper**
>
> > The paper appears hard to reproduce from the manuscript alone ... but specific implementation procedures and explanations related to experimental details are lacking.
>
> Thanks for the suggestion on reproducibility. The code of this paper had been uploaded along with the paper submission. Due to the page limitation of the main text, we put the details of the model implementation in the appendix (e.g., descriptions of the datasets and different configurations we used for experiments in Appendix C.1 and the details of model hyperparameters/architectures in Appendix C.2). In the first paragraph of the Experiments (Section 5), we quoted Appendix C.1 and Appendix C.2 to help readers understand the implementation details of models.

---

> > ### Comment · Reviewer_i5ZN · 2022-12-04
> > **Thank you for the detailed response.**
> >
> > This addresses most of my concerns. I am still leaning towards acceptance, but will defer to the other reviewers as my knowledge about the topic and related work appears to be insufficient to make an accurate decision that takes into account all the subtleties of this research sub-field.

---

### Author Response · Authors · 2022-11-30
**To all Reviewers**

Dear Reviewers,

Thanks for the constructive comments on our paper. We have uploaded the revised version of the paper, in which we attempted to solve these questions. And we hope our responses could address your concerns and would be happy to have further discussions if there are any other questions.

As the discussion period is nearing its end, don't hesitate to ask us if you have any questions on our responses.

Thanks, Authors

---

### Decision · Program_Chairs · 2023-01-20

**Decision:**

Accept: poster

**Justification For Why Not Higher Score:**

While compositionality is of wide interest, the current method doesn't scale well. Future variants of the method, if it can be scaled up and applied to more realistic problems, would merit far more exposure.

**Justification For Why Not Lower Score:**

The method is novel, the problem addressed is interesting and of interest to the community. It is a step forward and one that addresses compositionality in a generic way across many different problems.

**Metareview: Summary, Strengths And Weaknesses:**

Summary: Learn disentangled laws, rules that a domain observes, to reason about that domain in a compositional manner.
Strengths: The method is convincingly demonstrated across numerous domains. Compositionality is of wide interest to the community and this provides a novel way to encode and exploit it. The idea of learning generic laws rather than focusing on objects is important.
Weaknesses: Few baselines are compared against. Although reviewers could not point to a clear missing candidate method. Authors did add a GP-based method to alleviate this concern. The method is only applied to simple datasets and the current scaling properties of the method don't provide an obvious path forward. Reviewers found the background needed to understand the method was partially missing, this could be addressed in an expanded Appendix.


**Note From Pc:**

if the above contains the word "oral" or "spotlight" please see: "oral" presentation means -> notable-top-5% and "spotlight" means -> notable-top-25%. As stated in our emails, we are disassociating presentation type from AC recommendations